# GENERALIZED BELIEF TRANSPORT

## ABSTRACT

Human learners have ability to adopt appropriate learning approaches depending on constraints such as prior on the hypothesis and urgency of decision. However, existing learning models are typically considered individually rather than in relation to one and other. To build agents that have the ability to move between different modes of learning over time, it is important to understand how learning models are related as points in a broader space of possibilities. We introduce a mathematical framework, Generalized Belief Transport (GBT), that unifies and generalizes prior models, including Bayesian inference, cooperative communication and classification, as parameterizations of three learning constraints within Unbalanced Optimal Transport (UOT). We visualize the space of learning models encoded by GBT as a cube which includes classic learning models as special points. We derive critical properties of this parameterized space including proving continuity and differentiability which is the basis for model interpolation, and study limiting behavior of the parameters, which allows attaching learning models on the boundaries. Moreover, we investigate the long-run behavior of GBT, explore convergence properties of models in GBT mathematical and computationally, and formulate conjectures about general behavior. We conclude with open questions and implications for more unified models of learning.

## 1 INTRODUCTION

Learning and inference are subject to internal and external constraints. Internal constraints include the availability of relevant prior knowledge, which may be brought to bear on inferences based on data. External constraints include the availability of time to accumulate evidence or make the best decision now. Human learners appear to be capable of moving between these constraints as necessary. However, standard models of machine learning tend to view constraints as different problems, which impedes development of unified view of learning agents.

Indeed, internal and external constraints on learning map onto classic dichotomies in machine learning. Internal constraints such as the availability of prior knowledge maps onto the Frequentist-Bayesian dichotomy in which the latter uses prior knowledge as a constraint on posterior beliefs, while the former does not. Within Bayesian theory, a classic debate pertains to uninformative, or minimally informative, settings of priors (Jeffreys, 1946; Robert et al., 2009). External constraints such as availability of time to accumulate evidence versus the need to make the best possible decision now informs the use of generative versus discriminative approaches (Ng and Jordan, 2001). Despite the fundamental nature of these debates, and the usefulness of all approaches in the appropriate contexts, we are unaware of prior efforts to unify these perspectives and study the full space of possible models.

We introduce Generalized Belief Transport (GBT), based on Unbalanced Optimal Transport (Sec. 2), which paramterizes and interpolates between known reasoning modes (Sec. 3.2), with four major contributions. First, we prove continuity in the parameterization and differentiability on the interior of the parameter space (Sec. 3.1). Second, we analyze the behavior under variations in the parameter space (Sec. 3.3). Third, we consider sequential learning, where learners may (not) track the empirically observed data frequencies. And finally we state our theoretical results, simulations and conjectures about the sequential behaviors for various parameters for generic costs and priors (Sec. 4.2).

**Notations.** $\mathbb{R}_{\geq 0}$ denotes the non-negative reals. Vector $\mathbf{1} = (1, \ldots, 1)$. The $i$-th component of vector $v$ is $v(i)$. $\mathcal{P}(A)$ is the set of probability distributions over $A$. For a matrix $M$, $M_{ij}$ represents its $(i, j)$-th entry, $M_{(i,\_)}$ denotes its $i$-th row, and $M_{(\_,j)}$ denotes its $j$-th column. Probability is $\mathbb{P}(\,\cdot\,)$.

## 2    LEARNING AS A PROBLEM OF UNBALANCED OPTIMAL TRANSPORT

Consider a general learning setting: an agent, which we call a **learner**, updates their belief about the world based on observed data subject to constraints. There is a finite set $\mathcal{D} = \{d^1, \ldots, d^n\}$ of all possible data, that defines the interface between the learner and the world. The world is defined by a true hypothesis $h^*$, whose meaning is captured by a probability mapping $\mathbb{P}(d|h^*)$ onto observable data. For instance, the world can either be the environment in classic Bayesian inference (Murphy, 2012) or a **teacher** in cooperative communication (Wang et al., 2020b).

A learner is equipped with a set of hypotheses $\mathcal{H} = \{h^1, \ldots, h^m\}$ which may not contain $h^*$; an initial belief on the hypotheses set, denoted by $\theta_0 \in \mathcal{P}(\mathcal{H})$; and a non-negative cost matrix $C = (C_{ij})_{m \times n}$, where $C_{ij}$ measures the underlying cost of mapping $d^i$ into $h^j$ [1]. The cost matrix can be derived from other matrices that record the relation between $\mathcal{D}$ and $\mathcal{H}$, such as likelihood matrices in classic Bayesian inference or consistency matrices in cooperative communication (see details in Section 3.2).This setting reflects an agent's learning constraints: pre-selected hypotheses, and the relations between them and the communication interface (data set).

A learner observes data in sequence. At round $k$, the learner observes a data $d_k$ that is sampled from $\mathcal{D}$ by the world according to $\mathbb{P}(d|h^*)$. Then the learner updates their beliefs over $\mathcal{H}$ from $\theta_{k-1}$ to $\theta_k$ through a *learning scheme*, where $\theta_{k-1}, \theta_k \in \mathcal{P}(\mathcal{H})$. For instance, in Bayesian inference, the learning scheme is defined by Bayes rule; while in discriminative models, the learning scheme is prescribed by a code book.

The learner transforms the observed data into a belief on hypotheses $h \in \mathcal{H}$ with a minimal cost, subject to appropriate constraints, with the goal of learning the exact map $\mathbb{P}(d|h^*)$. We can naturally cast this learning problem as Unbalanced Optimal Transport.

### 2.1    UNBALANCED OPTIMAL TRANSPORT

Unbalanced optimal transport is a generalization of (entropic) Optimal Transport. Optimal transport infers a coupling that minimizes the cost of transporting between two marginal probability distributions (Monge, 1781; Kantorovich, 2006; Villani, 2008). Entropic Optimal Transport adds a regularization term based on the entropy of the inferred coupling, which has desirable computational consequences (Cuturi, 2013; Peyré and Cuturi, 2019). Unbalanced OT further relaxes the problem by allowing one to approximately match marginal probability distributions.

Let $\eta = (\eta(1), \ldots, \eta(n))$ and $\theta = (\theta(1), \ldots, \theta(m))$ be two probability distributions. A joint distribution matrix $P = (P_{ij})_{n \times m}$ is called a *transport plan* or *coupling* between $\eta$ and $\theta$ if $P$ has $\eta$ and $\theta$ as its marginals. Given a cost matrix $C = (C_{ij})_{n \times m} \in (\mathbb{R}_{\geq 0})^{m \times n}$, *Entropy regularized optimal transport (EOT)* (Cuturi, 2013) solves the optimal transport plan $P^{\epsilon_P}$ that minimizes the entropy regularized cost of transporting $\eta$ into $\theta$. Thus for a parameter $\epsilon_P > 0$: $P^{\epsilon_P} = \arg\min_{P \in U(\eta, \theta)} \langle C, P \rangle - \epsilon_P H(P)$, where $U(\eta, \theta)$ is the set of all transport plans between $\eta$ and $\theta$, $\langle C, P \rangle = \sum_{i,j} C_{ij} P_{ij}$ is the inner product between $C$ and $P$, and $H(P) = -\sum_{ij} P_{ij} \log P_{ij} + P_{ij}$ is the *entropy* of $P$.

**Unbalanced Optimal Transport (UOT)**, introduced by Liero et al. (2018), is a generalization of EOT that relaxes the marginal constraints. The degree of relaxation is controlled by two regularization terms. Formally, for non-negative scalar parameters $\epsilon = (\epsilon_P, \epsilon_\eta, \epsilon_\theta)$, the *UOT plan* is,

$$P^\epsilon(C, \eta, \theta) = \underset{P \in (\mathbb{R}_{\geq 0})^{n \times m}}{\arg\min} \quad \{\langle C, P \rangle - \epsilon_P H(P) + \epsilon_\eta \mathrm{KL}(P\mathbf{1}|\eta) + \epsilon_\theta \mathrm{KL}(P^T \mathbf{1}|\theta)\}. \quad (1)$$

Here $\mathrm{KL}(\mathbf{a}|\mathbf{b}) := \sum_i a_i \log(a_i/b_i) - a_i + b_i$ is the Kullback–Leibler divergence between vectors. UOT differs from EOT in relaxing the hard constraint that $P$ satisfy the given marginals $\eta$ and $\theta$, to soft constraints that penalize the marginals being far from $\eta$ or $\theta$ [2]. In particular, as $\epsilon_\eta$ and $\epsilon_\theta \to \infty$, we recover the EOT problem.

---

[1] To guarantee the hypotheses are distinguishable, we assume that $C$ does not contain two columns that are only differ by an additive scalar.

[2] UOT also generalizes to measures of arbitrary mass, i.e. the total mass of $\eta$ does not need to equal to $\theta$.

**Proposition 1.** *The UOT problem with cost matrix $C$, marginals $\theta, \eta$ and parameters $\epsilon = (\epsilon_P, \epsilon_\eta, \epsilon_\theta)$ generates the same UOT plan as the UOT problem with $tC$, $\theta$, $\eta$, $t\epsilon = (t\epsilon_P, t\epsilon_\eta, t\epsilon_\theta)$ for any $t \in (0, \infty)$. Therefore, the analysis on $\epsilon$ and $t\epsilon$ are the same for general cost $C$.*

The objective function in Eq. (1) is linear on $C, \epsilon_P, \epsilon_\eta, \epsilon_\theta$, so a positive common factor does not affect the solution. In the discussion under general cost matrix $C$, properties that hold for $\epsilon$ are also valid for all $t\epsilon$ ($t > 0$).

UOT plans can be solved efficiently via a Sinkhorn-style algorithm (Sinkhorn and Knopp, 1967). Roughly speaking, $(\eta, \theta, \epsilon)$-*unbalanced Sinkhorn scaling* of a matrix $M$ is iterated alternation of row and column normalizations of M with respect to $(\eta, \theta, \epsilon)$ (see Algorithm 2). Chizat et al. (2018) shows that: given a cost $C$, the UOT plan $P^\epsilon$ can be obtained by applying $(\eta, \theta, \epsilon)$-unbalanced Sinkhorn scaling on $K^\epsilon := e^{-\frac{1}{\epsilon_P}C} = (e^{-\frac{1}{\epsilon_P}C_{ij}})_{m \times n}$, with convergence rate $\tilde{\mathcal{O}}(\frac{mn}{\epsilon_P})$ (Pham et al., 2020).

**Generalized Belief Transport.** Learning, efficiently transport one's belief with constraints, is naturally a UOT problem. Each round, a learner, defined by a choice of $\epsilon = (\epsilon_P, \epsilon_\eta, \epsilon_\theta)$, updates their beliefs as follows. Let $\eta_{k-1}, \theta_{k-1}$ be the learner's estimations of the data distribution and the belief over hypotheses $\mathcal{H}$ after round $k - 1$, respectively. At round $k$, the learner first improves their estimation of the mapping between $\mathcal{D}$ and $\mathcal{H}$, denoted by $M_k$, through solving the UOT plan Eq. (1) with $(C, \eta_{k-1}, \theta_{k-1})$, i.e. $M_k = P^\epsilon(C, \eta_{k-1}, \theta_{k-1})$. Then with data observation $d_k$, the learner updates their beliefs over $\mathcal{H}$ using corresponding row of $M_k$, i.e. suppose $d_k = d^i$ for some $d^i \in \mathcal{D}$, the learner's belief $\theta_k$ is defined to be the row normalization of the $i$-th row of $M_k$. Finally, the learner updates their data distribution to $\eta_k$ by increment of the $i$-th element of $\eta_{k-1}$, see Algorithm 1.

---

**Algorithm 1** Generalized Belief Transport

  **input:** $C$, $\theta_0$, $\eta_0$, $h^*$, $N$, data sampler $\tau$ based on $\mathbb{P}(d|h^*)$, stopping condition $\omega$
  **output:** $M$, $\theta$
  **initialize:** $k \leftarrow 1$
  **while** $k < N$ **and not** $\omega(\theta)$ **do**

    $M \leftarrow P^\epsilon(C, \eta_{k-1}, \theta_{k-1})$
    get data $d^i$ sampled from $\tau$
    $\eta_k \leftarrow update(\eta_{k-1}, d^i)$ via update rule
    $\mathbf{v} \leftarrow M_{(i,\_)}$
    $\theta_k \leftarrow \mathbf{v}/\sum_{h \in \mathcal{H}} \mathbf{v}(h)$
    $k \leftarrow k + 1$
  **end while**

**Algorithm 2** Unbalanced Sinkhorn Scaling

  **input:** $C$, $\theta$, $\eta$, $\epsilon = (\epsilon_P, \epsilon_\eta, \epsilon_\theta)$, $N$ stopping condition $\omega$
  **output:** $P^\epsilon(C, \eta, \theta)$
  **initialize:** $\mathbf{K} = \exp(-\epsilon_P C)$, $\mathbf{v}^{(0)} = \mathbf{1}_m$
  **while** $k < N$ **and not** $\omega$ **do**

    $\mathbf{u}^{(k)} \leftarrow \left(\dfrac{\eta}{\mathbf{K}\mathbf{v}^{(k-1)}}\right)^{\frac{\epsilon_\eta}{\epsilon_\eta + \epsilon_P}}$,

    $\mathbf{v}^{(k)} \leftarrow \left(\dfrac{\theta}{\mathbf{K}^T\mathbf{u}^{(k)}}\right)^{\frac{\epsilon_\theta}{\epsilon_\theta + \epsilon_P}}$

  **end while**
  $P^\epsilon(C, \eta, \theta) = \mathrm{diag}(u)\mathbf{K}\mathrm{diag}(v)$

---

# 3 GENERALIZED BELIEF TRANSPORT (GBT)

Many learning models with different constraints—including Bayesian inference, Frequentist inference, Cooperative learning, and Discriminative learning—are unified under our GBT framework by varying the choice of $\epsilon$. In this section, we focus on the single-round behavior of the GBT model, i.e., given a pair of marginals $(\theta, \eta)$, how different learners update beliefs. We first visualize the entire learner set as a cube (in terms of parameters), see Figure 1. Then, we study the topological properties of the learner set through continuous deformations of parameters $\epsilon$. In particular, we show that existing models including Bayesian inference, cooperative inference and discriminative learning are learners with parameters $(1, 0, \infty)$, $(1, \infty, \infty)$ and $(0, \infty, \infty)$ respectively in our UOT framework.

## 3.1 THE PARAMETER SPACE OF GBT MODEL

The space of constrained belief-updating learners in GBT are parameterized by three regularizers for the underlying UOT problem (1): $\epsilon_P$, $\epsilon_\eta$ and $\epsilon_\theta$, each ranges in $[0, \infty)$. Therefore, the constraint space for GBT is $\mathbb{R}^3_{\geq 0}$, with the standard topology. When $C$, $\theta$ and $\eta$ are fixed (assume $\eta \in \mathbb{R}^m_+$), the map $\epsilon = (\epsilon_P, \epsilon_\eta, \epsilon_\theta) \mapsto (P^\epsilon)$ bears continuous properties:

**Proposition 2.** [3] *The UOT plan $P$ in Equation* (1)*, as a function of $\epsilon$, is continuous in $(0, \infty) \times [0, \infty)^2$. Furthermore, $P$ is differentiable with respect to $\epsilon$ in the interior.*

Continuity on $\epsilon$ provides the basis for interpolation between different learning agents. The proof of Proposition 2 also implies the continuity on $\eta$ and $\theta$. Further, towards the boundaries of the parameter space (where theories like Bayesian, Cooperative Communication live in), we show:

**Proposition 3.** *For any finite $s_P$, $s_\eta$, $s_\theta \geq 0$, the limit of $P^\epsilon$ exists as $\epsilon$ approaches $(\infty, s_\eta, s_\theta)$. In fact, $\lim_{\epsilon \to (\infty, s_\eta, s_\theta)} P_{ij}^\epsilon = 1$ for all $i, j$. Moreover, $P^\epsilon$ converges to the solution to*

$$\min \langle C, P \rangle - s_P H(P) + s_\theta KL(P^T \mathbf{1} | \theta), \text{ with constraint } P\mathbf{1} = \eta,$$

*as $\epsilon \to (s_P, \infty, s_\theta)$. Similarly, $P^\epsilon$ converges to the solution to*

$$\min \langle C, P \rangle - s_P H(P) + s_\eta KL(P\mathbf{1} | \eta), \text{ with constraint } P^T \mathbf{1} = \theta,$$

*as $\epsilon \to (s_P, s_\eta, \infty)$. And when $\epsilon \to (s_P, \infty, \infty)$, the matrix $P^\epsilon$ converges to the EOT solution:*

$$\min \langle C, P \rangle - s_P H(P), \text{ with constraints } P^T \mathbf{1} = \theta \text{ and } P\mathbf{1} = \eta.$$

*When $\epsilon \to (\infty, \infty, s_\theta), (\infty, s_\eta, \infty)$ or $(\infty, \infty, \infty)$, the limit does not exist, but the directional limits can be calculated.*

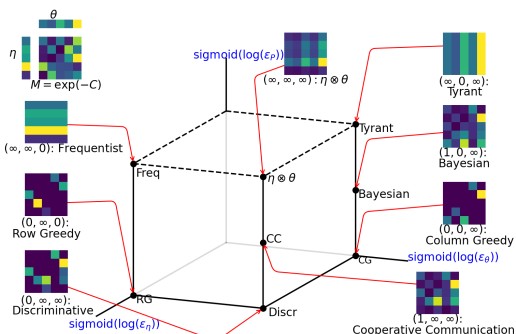

Figure 1: The parameter space $\mathcal{S}$ of GBT. Parameters $\epsilon = (\epsilon_P, \epsilon_\eta, \epsilon_\theta)$ can take the value $\infty$, rendering the corresponding regularization to a strict constraint. The two dashed edges with $\epsilon_P = \infty$ are not generally well-defined since the limits do not exist. The vertices corresponding to $\theta \otimes \eta$, Frequentist ($\eta \otimes \mathbf{1}$) and $\mathbf{1} \otimes \theta$ are the limits taken along the vertical edges. Given $(C, \theta, \eta)$ as shown in the left corner, each colored map plots each GBT learner (differ by constraints)'s estimation of the mapping between hypotheses and data (UOT plan).

The generalized parameter space for UOT with its boundaries can be visualized in Fig. 1. Function sigmoid$(\log(x))$ maps segment $[0, \infty)$ to $[0, 1)$ smoothly. Then we can add boundaries to the image cube $[0, 1)^3$. The dashed lines in the figure indicates limits that do not exist. The parameter space is then $\mathcal{S} = [0, \infty]^3 \setminus (\{(\infty, \infty, x) : x \in [0, \infty]\} \cup \{(\infty, x, \infty) : x \in [0, \infty]\})$. Later, we may still mention $(\infty, \infty, \epsilon_\theta)$ and $(\infty, \epsilon_\eta, \infty)$, only for case where the direction is vertical (along axis of $\epsilon_P$).

### 3.2 SOME SPECIAL POINTS IN THE PARAMETER SPACE

**Bayesian Inference.** Given a data observation, a Bayesian learner (BI) (Murphy, 2012) derives posterior belief $\mathbb{P}(h|d)$ based on prior belief $\mathbb{P}(h)$ and likelihood matrix $\mathbb{P}(d|h)$, according to the Bayes rule. Intuitively, due to soft time constraint ($\epsilon_P = 1$), a Bayesian learner is a generative agent who puts a hard internal constraint on their prior belief ($\epsilon_\theta = \infty$), and omits the estimated data distribution $\eta$ in the learning process, ($\epsilon_\eta = 0$). As a direct application of Prop 3, we show:

**Corollary 4.** *Consider a UOT problem with cost $C = -\log \mathbb{P}(d|h)$, marginals $\theta = \mathbb{P}(h)$, $\eta \in \mathcal{P}(\mathcal{D})$. The optimal UOT plan $P^{(1, \epsilon_\eta, \epsilon_\theta)}$ converges to the posterior $\mathbb{P}(h|d)$ as $\epsilon_\eta \to 0$ and $\epsilon_\theta \to \infty$. Bayesian inference is a special case of GBT with $\epsilon = (1, 0, \infty)$.*

Moreover, by relaxing the constraint on the prior (i.e., $0 < \epsilon_\theta < \infty$), one obtains a parameterized family of less informative priors.

---

[3]Proofs of all claims are included in the Appendices.

**Frequentist Inference.** A frequentist updates their belief from data observations by increasing the corresponding frequencies of datum. Intuitively, a frequentist is an agent who puts a hard constraint on the data distribution $\eta$ ($\epsilon_\eta = \infty$), and omits prior knowledge $\theta$ ($\epsilon_\theta = 0$) in a learning process without time constraint ($\epsilon_P = \infty$). Formally we show:

**Corollary 5.** *Consider a UOT problem with $\theta \in \mathcal{P}(\mathcal{H})$, $\eta = \mathbb{P}(d)$. The optimal UOT plan $P^{(\epsilon_P, \infty, 0)}$ converges to $\eta \otimes \mathbf{1}$ as $\epsilon_P \to \infty$. Frequentist Inference is a special case of GBT with $\epsilon = (\infty, \infty, 0)$.*

**Cooperative Communication.** Two cooperative agents, a teacher and a learner, are considered in Yang et al. (2018); Wang et al. (2020b); Shafto et al. (2021). Cooperative learners (CI) draw inferences about hypotheses based on which data would be most effective for the teacher to choose (see a brief model summary in the Appendix A). Given a data observation, a cooperative learner derives an optimal plan $L = \mathbb{P}(\mathcal{H}, \mathcal{D})$ based on a prior belief $\mathbb{P}(h)$, a shared data distribution $\mathbb{P}(d)$ and a matrix $M$ specifies the consistency between data and hypotheses (such as $M_{ij}$ records the co-occurrence of $d^i$ and $h^j$). Intuitively, a cooperative learner is also a generative agent who puts hard constraints on both data and hypotheses ($\epsilon_\eta = \infty, \epsilon_\theta = \infty$), and aims to align with the true belief asymptotically, ($\epsilon_P = 1$). Thus as a direct application of Proposition 3 we show:

**Corollary 6.** *Let cost $C = -\log M$, marginals $\theta = \mathbb{P}(h)$ and $\eta = \mathbb{P}(d)$. The optimal UOT plan $P^{(1, \epsilon_\eta, \epsilon_\theta)}$ converges to the optimal plan $L$ as $\epsilon_\eta \to \infty$ and $\epsilon_\theta \to \infty$. Cooperative Inference is a special case of GBT with $\epsilon = (1, \infty, \infty)$, which is exactly entropic Optimal Transport (Cuturi, 2013).*

**Discriminative learning.** A discriminative learner decodes an uncertain, possibly noise corrupted, encoded message, which is a natural bridge to information theory (Cover, 1999; Wang et al., 2020b). A discriminative learner builds an optimal map to hypotheses $\mathcal{H}$ conditioned on observed data $\mathcal{D}$. The map is perfect when, for all messages, encodings are uniquely and correctly decoded. Intuitively, a discriminative learner aims to quickly build a deterministic code book (implies $\epsilon_P = 0$) that matches the marginals on $\mathcal{H}$ and $\mathcal{D}$. Thus, discriminative learner is GBT with $\epsilon = (0, \infty, \infty)$:

**Corollary 7.** *Consider a UOT problem with cost $C = -\log \mathbb{P}(d, h)$, $m = n$, and marginals $\theta = \eta$ are uniform. The optimal UOT plan $P^{(\epsilon_P, \epsilon_\eta, \epsilon_\theta)}$ approaches to a diagonal matrix as $\epsilon_\eta, \epsilon_\theta \to \infty$ and $\epsilon_P \to 0$. In particular, discriminative learner is a special case of GBT with $\epsilon = (0, \infty, \infty)$, which is exactly classical Optimal Transport (Villani, 2008).*

Many other interesting models are unified under GBT framework as well. GBT with $\epsilon = (0, \infty, 0)$ denotes Row Greedy learner which is widely used in Reinforcement learning community (Sutton and Barto, 2018); $\epsilon = (\infty, \infty, \infty)$ yields $\eta \otimes \theta$ which is independent coupling used in $\chi^2$ (Fienberg et al., 1970); $\epsilon = (\epsilon_P, \epsilon_\theta, \infty)$ is used for adaptive color transfer studied in (Rabin et al., 2014); and $\epsilon = (0, \epsilon_\theta, \epsilon_\eta)$ is UOT without entropy regularizer developed in (Chapel et al., 2021). Other points in the GBT parameter space are also of likely interest, past or future.

### 3.3 GENERAL PROPERTIES ON THE TRANSPORTATION PLANS

The general GBT framework builds a connection between the above theories, and the behavior of theory varies according to the change of parameters. In particular, each factor of $\epsilon = (\epsilon_P, \epsilon_\eta, \epsilon_\theta)$ expresses different constraints of the learner. Given $(C, \theta, \eta)$ as shown in the top-left corner of Fig. 1, we plot each learner's UOT plan with darker color representing larger elements.

$\epsilon_P$ controls a learner's learning horizon. When $\epsilon_P \to 0$, agents are under the time pressure of making immediate decision, hence GBT converges a discriminative learner, or Row Greedy learner on the bottom of the cube (Fig. 1). Their UOT plans have a clear leading diagonal which allows them to make fast decisions. Most of the time, one datum is enough to identify the true hypothesis and convergence is achieved within every data observation. When $\epsilon_P \to \infty$, GBT converges to a reticent learner, such as learners on the top of the cube. Data do not constrain the true hypothesis, and learners draw their conclusions independent of the data. In between, GBT provides a generative (probabilistic) learner. When $\epsilon_P = 1$, we have Bayesian learner and Cooperative learner, for whom data accumulate to identify the true hypothesis in a manner broadly consistent with probabilistic inference, and consistency is asymptotic.

$\epsilon_\eta$ controls a learner's knowledge on the data distribution $\eta$. When $\epsilon_\eta \to \infty$, GBT converges to a learner who is aware of the data distribution and reasons about the observed data according to the probabilities/costs of possible outcomes. Examples include the Discriminative and Cooperative

learners on the front of the cube. When $\epsilon_\eta \to 0$, GBT converges to a learner who updates their belief without taking $\eta$ into consideration, such as Bayesian learners on the back of the cube, and the Tyrant who does not care about data nor cost and is impossible to be changed by anybody.

$\epsilon_\theta$ controls the strength of availability of prior knowledge for the learner. When $\epsilon_\theta \to \infty$, GBT converges to a learner who enforces a prior over the hypotheses, such as Bayesian, Cooperative and Discriminative learners on the right of the cube. Actually, BGT follows Bayes rule when $\epsilon_\theta = \infty$ (Prop 8). When $\epsilon_\theta \to 0$, GBT converges to learners who utilizes no prior knowledge. Hence they do NOT maintain beliefs over $\mathcal{H}$, and draws their conclusions purely on the data distribution, such as a Frequentist learner $\eta \otimes \mathbf{1}$ on the left of the cube.

**Proposition 8.** *In GBT with $\epsilon_\theta = \infty$, cost $C$ and current belief $\theta$. The learner updates $\theta$ with UOT plan in the same way as applying Bayes rule with likelihood from $P^\epsilon(C, \eta, \theta)$, and prior $\theta$.*

## 4 SEQUENTIAL GBT: ASYMPTOTIC BEHAVIOR

One interesting difference between the one-shot case considered above and the sequential case is the possibility of observing many data points. In addition to the learning models in the GBT parameter space, in this section, we consider whether the learner's marginal on the data is fixed a priori, or accumulates evidence based on experience.

### 4.1 BASICS

The sequential GBT model consists of a teacher and a learner. The teacher samples data from a probability distribution $\eta$ (not necessarily related to some $h \in \mathcal{H}$), and the learner follows GBT with cost $C$, and parameter $\epsilon$. The learner starts with a prior $\theta_0$, and applies in each round $k$ GBT with $\eta_{k-1}$ and $\theta_{k-1}$ to generate $\theta_k$ through the UOT solution $M_k$. In the Preliminary sequential model (**PS**), we assume $\eta_k = \eta$ for all $k$. However, in practice, a learner does not have access $\eta = \mathbb{P}(d|h^*)$. Instead, in each round the learner may choose to use the current statistical distribution in data as an estimation of $\eta$, i.e., $\eta_k(d) \propto |\{i : i < k, d_i = d\}| + n_0(d)$ according to the observed data sequence, where $n_0(d) > 0$ (e.g., 1 as in add-one smoothing (Murphy, 2012)) is the prior counts to avoid zero in $\eta$. Thus we have the Real sequential model (**RS**) where $\eta_k \xrightarrow{a.s.} \eta$. It is easy to see that the sequence of posteriors form a time-homogeneous Markov chain on $\mathcal{P}(\mathcal{H})$.

In statistics, a model is said to be consistent (strongly-consistent) when, for every fixed hypothesis $h \in \mathcal{H}$, the model's belief $\theta$ over the hypotheses set $\mathcal{H}$ converges to $\delta_h$ in probability (almost surely) as more and more data are sampled from $\eta = \mathbb{P}(d|h)$, when $\theta$'s are considered random variables. The consistency has been well studied for Bayesian Inference since Bernstein and von Mises and Doob (Doob, 1949), and recently demonstrated for Cooperative Communication (Wang et al., 2020a). The challenging problem arises when one tries to learn a $h^*$ that is not contained in the pre-selected hypothesis space $\mathcal{H}$. It is not clear which $h \in \mathcal{H}$ is the 'correct' target to converge to. Thus consistency does not fit the situation in sequential GBT.

For sequential GBT models, we state the properties directly in the language of posterior sequence $(\Theta_k)_{k=1}^\infty$ as random variables, and name them if necessary. We focus on whether the sequence converges (and in which sense), and how conclusive (how likely to provide a stable, fixed $h \in \mathcal{H}$ as the result) the sequence is. We provide some theoretical conclusions, and fill the gaps with empirical results and conjectures.

### 4.2 RESULTS AND CONJECTURES

Results in this section are stated on different $\epsilon_\theta$ values. According to Prop. 1, we could focus on $\epsilon_P = 1$ for generic cost matrix $C$, and general result of $(\epsilon_P, \epsilon_\eta, \epsilon_\theta)$ becomes the same as the $(1, \epsilon_\eta/\epsilon_P, \epsilon_\theta/\epsilon_P)$ case. So we choose $\epsilon_P = 1$ in simulations.

$\epsilon_\theta = \infty$**: Conclusive and Bayesian-style.** These are located on the right side of Fig. 1, and contain many well-studied learners: Bayesian, Cooperative, Discriminative, Row Greedy etc. According to Prop 8, learners in this class perform "Bayesian" style learning.

There are two theoretical results: $\epsilon_\eta = 0$ (Bayesian) and $\epsilon_\eta = \infty$ (SCBI learner (Wang et al., 2020a)). Others are explored in simulations.

**Theorem 9** ((Doob, 1949),(Wang et al., 2020a)). *In GBT sequential model (both (**PS**) and (**RS**)) with $\epsilon = (\epsilon_P, 0, \infty)$ where $\epsilon_P \in (0, \infty)$, the sequence $\Theta_k$ converges to some $\delta_h$ almost surely, $h$ is the closest column of $e^{-C/\epsilon_P}$ to $\eta$ in the sense of KL-divergence, when $\theta$ is positive (on each entry).*

When $\epsilon_\eta = \epsilon_\theta = \infty$, the models (**PS**) and (**RS**) have slightly different behaviors.

**Lemma 10.** For $\epsilon = (\epsilon_P, \infty, \infty)$, $\epsilon_P \in (0, \infty)$, given cost $C$ with initial belief $\theta_0 \in \mathcal{P}(\mathcal{H})$ and fixed teaching and learning distribution $\eta_k = \eta \in \mathcal{P}(\mathcal{D})$ for all $k$, i.e., model (**PS**), then the belief random variables $(\Theta_k)_{k \in \mathbb{N}}$ have the same expectation on $h$: $\mathbb{E}_{\Theta_k}[\theta(h)] = \theta_0(h)$.

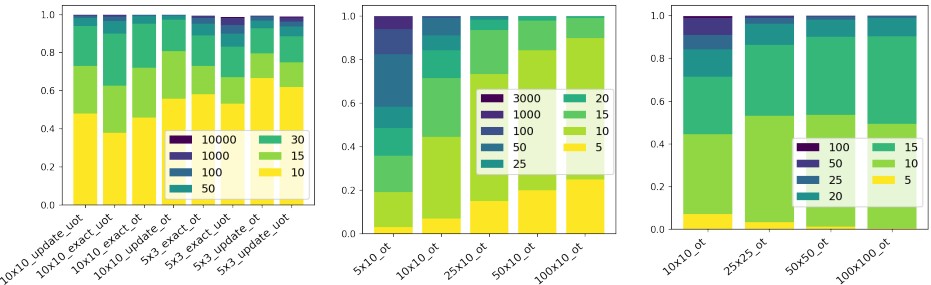

Figure 2: Evidence of general consistency: we plot the percentage of episodes that reaches a threshold (0.999) by round number (in colors of the bars). Each bar represents a size of matrix, for each bar 100 matrices were randomly sampled, and 1000 rounds were simulated per matrix. "exact" means learner uses $\eta_k = \eta$, (**PS**), "update" means learner uses statistics on current data in the episode (**RS**). "uot" takes $\epsilon = (1, 40, 40)$ and "ot" comes with exact and $\epsilon = (1, \infty, \infty)$.

**Theorem 11 (PS).** *Consider a learning problem with initial belief $\theta_0 \in \mathcal{P}(\mathcal{H})$, and the true hypothesis $h^*$ defined by $\eta \in \mathcal{P}(\mathcal{D})$. If the learner's data distribution $\eta_k = \eta$, then belief random variables $(\Theta_k)_{k \in \mathbb{N}}$ converge to the random variable $Y$ in probability, where $Y = \sum_{h \in \mathcal{H}} \theta_0(h)\delta_h$ and $Y$ is supported on $\{\delta_h\}_{h \in \mathcal{H}}$ with $\mathbb{P}(Y = \delta_h) = \theta_0(h)$ for $\epsilon_\eta = \epsilon_\theta = \infty$ and $\epsilon_P \in (0, \infty)$.*

**Corollary 12.** *Given a fixed data sequence $d_i$ sampled from $\eta$, if $\theta_k$ converges to $\delta_{h^j}$, then the $j$-th column of $M_k$ converges to $\eta$.*

Thus a GBT learner, with access to the data distribution and using strict marginal constraints, converges to a distribution on $\mathcal{D}$ same as $\eta$ with probability 1. Moreover, the probability of which column $h$ is shaped into $\eta$ is determined by their prior $\theta_0$. That is, GBT learners converge to the truth by changing one of their original hypotheses into the true hypothesis.

For the (**RS**) model, the result is similar, but Lemma 10 fails to hold:

**Proposition 13.** *Consider a learning problem with cost $C$, initial belief $\theta_0 \in \mathcal{P}(\mathcal{H})$, the true hypothesis $h^*$ defined by $\eta \in \mathcal{P}(\mathcal{D})$. For the (**RS**) problem, the belief random variables $(\Theta_k)_{k \in \mathbb{N}}$ satisfies that for any $s > 0$, $\lim_{k \to \infty} \sum_{h \in \mathcal{H}} \mathbb{P}(\Theta(h) > 1 - s) = 1$. As a consequence, $M_k$ as the transport plan has a dominant column ($h^j$) with total weights $> 1 - s$, and $|(M_k)_{ij} - \eta_k(i)| < s$.*

In fact, as long as the sequence of $\eta_k$ as random variables converges to $\eta$ in probability, the above proposition holds. The limit $\lim_{k \to \infty} \sum_{h \in \mathcal{H}} \mathbb{P}(\Theta(h) > 1 - s)$ measures how conclusive the model is.

In contrast with standard Bayesian or other inductive learners, Proposition 13 shows that a GBT learner is able to learn *any* hypothesis mapping $\eta = \mathbb{P}(d|h^*)$ up to a given threshold $s$ with probability 1. In addition to unifying disparate models of learning, GBT enables a fundamentally more powerful approach to learning by empirically monitoring the data marginal.

Fig. 2 illustrates convergence over learning problems and episodes. In each bar, we sample 100 learning problem $(C, \theta_0, h^*)$ from Dirichlet distribution with hyperparameters the vector $\mathbf{1}$. Then we sample 1000 data sequences (episodes) of maximal length $N = 10000$. The learner learns with Algorithm 1 where the stopping condition $\omega$ is set to be $\max_{h \in \mathcal{H}} \theta(h) > 1 - s$ with $s = 0.001$. The $y$-axis in the plots represents the percentage of total episode converged. The color indicates in how many rounds the episode converges. For instance, in the bar corresponding to '$10 \times 10$\_update\_uot', with 10 data points (yellow portion), about $50\%$ episodes satisfy the stopping condition.

In Figure 2, the first plot shows results for $10 \times 10$ and $5 \times 3$ matrices. The second plot shows results for rectangular matrices of dimension $m \times 10$ with $m$ ranges in $[5, 10, 25, 50, 100]$. The third plot shows results for square matrices of dimension $m \times m$ with $m$ ranges in $[10, 25, 50, 100]$. Here 'exact' and 'update' indicate the problem is (**PS**) or (**RS**), respectively. For parameters, $uot$ represents the parameter choice $(\epsilon_P = 1, \epsilon_\theta = \epsilon_\eta = 40)$ vs. $ot$ represents the parameter choice $(\epsilon_P = 1, \epsilon_\theta = \epsilon_\eta = \infty)$. The first plot illustrates that learners that do not have access to the true hypothesis (empirically builds estimation of $\eta$) learn faster than learners who have full access. The second plot indicates with a fixed number of hypotheses, learning is faster when the dimension of $\mathcal{D}$ increases. The third plot shows that the GBT learner scales well with the dimension of the problem.

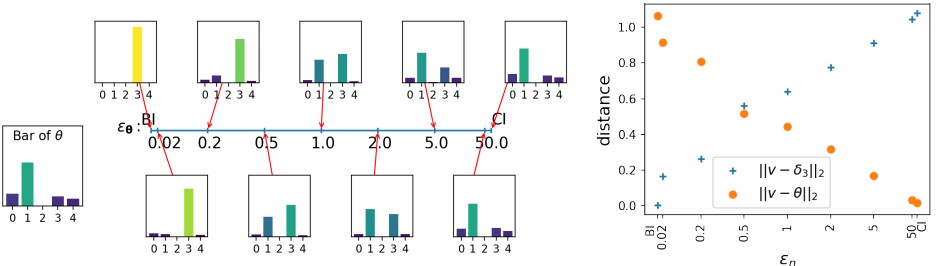

Figure 3: Left: Behavior of models spanning the line segment between BI and CI. With $\epsilon_P = 1$ and $\epsilon_\theta = \infty$, when $\epsilon_\eta$ varies from 0 to $\infty$, the theory changes from BI to CI. Each bar graphs the Monte-Carlo result of $400,000$ teaching sequences, we empirically observe that the coefficients $a(h)$ of the limit in terms of $\sum_{h \in \mathcal{H}} a(h)\delta_h$ changes from BI to CI continuously from $\delta(h^3)$ by Bernstein-von Mises to $\theta_0(h)$ by Theorem 11. Right: the Euclidean distances of each coefficient $a(h)$ to BI result (blue crosses), and to CI result (orange dots).

Then we study the learners that interpolate between Bayesian and Cooperative learners (located on the line connecting CI and BI in Fig 1). Consider a fixed learning problem $(C, \theta_0, h^*)$. Consistency of Bayesian inference states that asymptotically, the learner Bayesian converges to a particular hypothesis $h_b \in \mathcal{H}$ almost surely where $h_b$ is the hypothesis closest to $h^*$ under KL divergence. Theorem 11 indicates that a GBT cooperative learner modifies one of the hypotheses into $h^*$ in probability 1. The probability of $h^j$ converges to $h^*$ is determined by $\theta_0(h^j)$.

In Fig. 3, we study the asymptotic behavior of the learners corresponding to $\epsilon = (1, \epsilon_\eta, \infty)$, with $\epsilon_\eta \in \{0, 0.02, 0.2, 0.5, 1, 2, 5, 50, \infty\}$. We sample a learning problem with a dimension $5 \times 5$ from Dirichlet distribution with hyperparameters the vector $\mathbf{1}$. Each learner $\epsilon = (1, \epsilon_\eta, \infty)$ is equipped with a fixed $C$, $\theta_0$ and $\eta_k = \eta$ for all $k$. We run $400,000$ learning episodes per learner, and plot their convergence summary in the bar graph. A continuous transition from a Bayesian learner to a cooperative learner can be empirically observed: the coefficients $a(h)$ of the limit in terms of $\sum_{h \in \mathcal{H}} a(h)\delta_h$ changes from $\delta(h^3)$ by Bernstein-von Mises to $\theta_0(h)$ by Theorem 11.

From the previous empirical results, we conclude the following conjecture:

**Conjecture 14.** *When $\epsilon = (\epsilon_P, \epsilon_\eta, \infty)$, where $\epsilon_P \in (0, \infty)$, the sequence of posteriors $\Theta_k$ from generic $C$, $\eta$, $\theta$ and $\epsilon$ as random variables satisfy $\lim_{k \to \infty} \sum_{h \in \mathcal{H}} \mathbb{P}(|\Theta_k(h) - 1| < e) = 1$ for any $e > 0$.*

We further report an empirical property observed in simulation, which suggests a possible rate of convergence. For given $C$, $\theta_0$ and $\eta$, fix $\epsilon_P = 1$ and $\epsilon_\theta = \infty$, as $\epsilon_\eta$ changes from 0 to $\infty$, we pick out those episodes with $\theta_N(h) > 0.95$ and plot the values $\mathbb{E}_{\theta_N(h) > 0.95}[\ln \theta_k(h) - \ln(1 - \theta_k(h))]$ for each $h$ against $k$ (Fig. 4 bottom). Near linear relations are observed away from the first several rounds and before the values reaches the precision threshold. These are empirical estimates of the rate of convergence.

There is a special case on the boundary, the Independent Coupling $(\infty, \infty, \infty)$, whose limit is taken vertically along $\epsilon_P$-axis, see Sec. 3.1. Independent Coupling has a fixed posterior, where $Law(\Theta_k) = \delta_{\theta_0}$, as the normalization of each row of $P^{(\infty, \infty, \infty)}$ is $\theta_0$.

$\epsilon_\theta = 0$: **Inconclusive and independent.** The following holds for both (**PS**) and (**RS**):

**Proposition 15.** *For $\epsilon = (\epsilon_P, \epsilon_\eta, 0)$ with $\epsilon_P \in (0, \infty)$, as $\eta_k \to \eta$ almost surely, the sequence $\Theta_k$ of posteriors as a sequence of random variables converges in probability to variable $\Theta$, where*

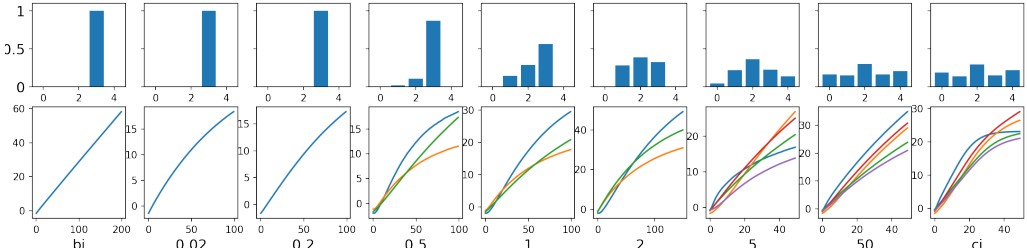

Figure 4: Top: For a learning problem $C$, behaviors of 9 different learners with $\epsilon_P = 1$, $\epsilon_\theta = \infty$ and various $\epsilon_\eta$ (denoted in figure) on conclusion distributions, $a(h)$ in bar graph, plots below bars are estimated convergence rates $\mathbb{E} \ln(\theta_k(h)/(1 - \theta_k(h)))$ averaged on episodes converging to $h$, one curve per hypothesis.

$\mathbb{P}(\Theta = \mathbf{v}^i) = \eta(i)$ *and* $\mathbf{v}^i = P_{(i,\_)}/\left(\sum_{j=1}^m P_{ij}\right)$ *and* $P = P^\epsilon(C, \eta, \theta)$. *Therefore, for any $s > 0$,* $\lim_{k\to\infty} \sum_{h\in\mathcal{H}} \mathbb{P}(|\Theta_k(h) - 1| < s) = 0$ *for generic (for all but in a closed subset) cost $C$ and $\eta$, $\theta$.*

With $\epsilon_\theta = 0$, the constraint on column-sum ($\epsilon_\eta$-term) fails to affect the transport plan, thus the $\Theta_k$'s in the sequence are independent from each other, in contrast that in all other cases the adjacent ones are correlated via a nondegenerate transition distribution. The independence makes the sequence of posterior-samples in one episode behave totally random, thus rarely converge as points in $\mathcal{P}(\mathcal{H})$. Furthermore, when consider the natural coupling $(\Theta_{k-1}, \Theta_k)$ from Markov transition measure for $\epsilon_\theta = 0$ (which is independent), $\mathbb{E}\left(|\Theta_{k-1} - \Theta_k|^2\right)$ converges to the variance $Var(\eta)$. In contrast, for $\epsilon_\theta = \infty$, $\mathbb{E}\left(|\Theta_{k-1} - \Theta_k|^2\right)$ converges to 0 if Conj. 14 holds.

$\epsilon_\theta \in (0, \infty)$: **partially conclusive.** From Conj. 14 and Prop. 15, together with the continuity of the transition distribution on $\epsilon$, we conjecture the following continuity on conclusiveness when $\epsilon_P \in (0, \infty)$.

**Conjecture 16.** *For both (PS) and (RS) models, when $\epsilon = (\epsilon_P, \epsilon_\eta, \epsilon_\theta)$ with $\epsilon_P, \epsilon_\theta \in (0, \infty)$, the posterior sequence $\Theta_k$ from generated from generic $C$, $\eta$, $\theta$ and $\epsilon$ satisfy that $\lim_{k\to\infty} \sum_{h\in\mathcal{H}} \mathbb{P}(|\Theta_k(h) - 1| < s) = L$ exists, and $L \in (0, 1)$, for any $s > 0$.*

## 5 RELATED WORK

Prior work defines and outlines basic properties of Unbalanced Optimal Transport (Liero et al., 2018; Chizat et al., 2018; Pham et al., 2020). Bayesian approaches are prominent in machine learning (Murphy, 2012) and beyond (Jaynes, 2003; Gelman et al., 1995). There is also research on cooperative learning (Wang et al., 2019; 2020b;a) see also (Liu et al., 2021; Yuan et al., 2021; Zhu, 2015; Liu et al., 2017; Shafto and Goodman, 2008; Shafto et al., 2014; Frank and Goodman, 2012; Goodman and Frank, 2016; Fisac et al., 2017; Ho et al., 2018; Laskey et al., 2017). Discriminative learning is the reciprocal problem in which one sees data and asks which hypothesis best explains it (Ng and Jordan, 2001; Mandler, 1980). We are unaware of any work that attempts to unify and analyze the general problem of learning in which each of these are instances.

## 6 CONCLUSIONS

We have introduced Generalized Belief Transport (GBT), which unifies and parameterizes classic instances of learning including Bayesian inference, Cooperative Inference, and Discrimination, as Unbalanced Optimal Transport (UOT). We show that each instance is a point in a continuous, differentiable on the interior, 3-dimensional space defined by the regularization parameters of UOT. In addition to supporting generalized learning, we prove and illustrate asymptotic consistency and estimate rates of convergence, including convergence to hypotheses with zero prior support. In summary, GBT unifies very different modes of learning, yielding a powerful, general framework for modeling learning agents.

ETHIC STATEMENT

The main contributions of this paper are theoretical, rather than practical, in nature. While understanding learning and inference in more unified and generalized ways may have broad impact including causing any ethic problems, nothing is likely to be realized as a direct consequence of this work.

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

## A  ADDITIONAL MATERIALS

---
**Algorithm 3** Unbalanced Sinkhorn Scaling
---

 **input:** $C$, $\theta$, $\eta$, $\epsilon = (\epsilon_P, \epsilon_\eta, \epsilon_\theta)$, $N$ stopping condition $\omega$
 **initialize:** $\mathbf{K} = \exp(-\epsilon_P C)$, $\mathbf{v}^{(0)} = \mathbf{1}_m$
 **while** $k < N$ **and not** $\omega$ **do**
   $\mathbf{u}^{(k)} \leftarrow \left(\frac{\eta}{K\mathbf{v}^{(k-1)}}\right)^{\frac{\epsilon_\eta}{\epsilon_\eta + \epsilon_P}}$, $\mathbf{v}^{(k)} \leftarrow \left(\frac{\theta}{K^T\mathbf{u}^{(k)}}\right)^{\frac{\epsilon_\theta}{\epsilon_\theta + \epsilon_P}}$
 **end while**
 **output:** $M = \text{diag}(u) K \text{diag}(v)$

---

**Cooperative Communication.**  *Cooperative communication* formalizes a single problem comprised of interactions between two processes: teaching and learning. The teacher and learner have beliefs about hypotheses, which are represented as probability distributions. The process of teaching is to select data that move the learner's beliefs from some initial state, to a final desired state. The process of learning is then, given the data selected by the teacher, infer the beliefs of the teacher. The teacher's selection and learner's inference incur costs. The agents minimize the cost to achieve their goals. Communication is successful when the learner's belief, given the teacher's data, is moved to the target distribution.

Formally, denote the common ground between agents: the shared priors on $\mathcal{H}$ and $\mathcal{D}$ by $\mathbb{P}(h)$ and $\mathbb{P}(d)$, the shared initial matrix over $\mathcal{D}$ and $\mathcal{H}$ by $M$ of size $|\mathcal{D}| \times |\mathcal{H}|$. In general, up to normalization, $M$ is simply a non-negative matrix which also specifies the consistency between data and hypotheses[4]

In cooperative communication, a learner's goal is to minimize the cost of transforming the observed data distribution $\mathbb{P}(\mathcal{D})$ to the shared prior over hypotheses $\mathbb{P}(\mathcal{H})$. A learner's cost matrix $C^L = (C^L_{ij})_{|\mathcal{M}| \times |\mathcal{H}|}$ is defined as $C^L_{ij} = -\log M$. A *learning plan* is a joint distribution $L = (L_{ij})$, where $L_{ij} = P_L(d_i, h_j)$ represents the probability of the learner inferring $h_j$ given $d_i$. It is proved in (Wang et al., 2019) that:

**Proposition 17.** *Optimal cooperative communication plans, L, is the EOT plan with cost $C^L$ and marginals being $\eta = \mathbb{P}(d)$ and $\theta = \mathbb{P}(h)$.*

## B  PROOFS

**Proposition 1.** *The UOT problem with cost matrix $C$, marginals $\theta, \eta$ and parameters $\epsilon = (\epsilon_P, \epsilon_\eta, \epsilon_\theta)$ generates the same UOT plan as the UOT problem with $tC$, $\theta$, $\eta$, $t\epsilon = (t\epsilon_P, t\epsilon_\eta, t\epsilon_\theta)$ for any $t \in (0, \infty)$.*

*Proof.*  Consider that the UOT problem solution is

$$P^\epsilon(C, \eta, \theta) = \underset{P \in (\mathbb{R}_{\geq 0})^{n \times m}}{\arg\min} \quad \{\langle C, P \rangle - \epsilon_P H(P) + \epsilon_\eta \text{KL}(P\mathbf{1}|\eta) + \epsilon_\theta \text{KL}(P^T\mathbf{1}|\theta)\}. \quad (2)$$

where the objective function is linear on $C$ and $\epsilon$.

$$
\begin{aligned}
P^{t\epsilon}(tC, \eta, \theta) &= \underset{P \in (\mathbb{R}_{\geq 0})^{n \times m}}{\arg\min} \quad \{\langle tC, P \rangle - t\epsilon_P H(P) + t\epsilon_\eta \text{KL}(P\mathbf{1}|\eta) + t\epsilon_\theta \text{KL}(P^T\mathbf{1}|\theta)\} \\
&= \underset{P \in (\mathbb{R}_{\geq 0})^{n \times m}}{\arg\min} \quad t \cdot \{\langle C, P \rangle - \epsilon_P H(P) + \epsilon_\eta \text{KL}(P\mathbf{1}|\eta) + \epsilon_\theta \text{KL}(P^T\mathbf{1}|\theta)\} \\
&= P^\epsilon(C, \eta, \theta). \quad (3)
\end{aligned}
$$

$\square$

**Proposition 2.** *The UOT plan $P$ in Equation (1), as a function of $\epsilon$, is continuous in $(0, \infty) \times [0, \infty)^2$. Furthermore, $P$ is differentiable with respect to $\epsilon$ in the interior.*

---
[4]Data, $d_i$, are consistent with a hypothesis, $h_j$, when $M_{ij} > 0$.

*Proof.* For simplicity, in this proof, for a vector $v$, we use both $v_i$ and $v(i)$ to represent a component of $v$.

By definition, the UOT plan $P$ minimizes the objective function $\Omega(P; \epsilon) = \langle C, P \rangle - \epsilon_P H(P) + \epsilon_\eta KL(P\mathbf{1}|\eta) + \epsilon_\theta KL(P^T\mathbf{1}|\theta)$. Since $\Omega$ is a strict convex function on $P$, there is only one minimal $P$. So the UOT plan $P$ is the solution to $\nabla_P \Omega = 0$. From a direct calculation,

$$(\nabla_P \Omega)_{ij} = C_{ij} + \epsilon_P \ln P_{ij} + \epsilon_\eta (\ln(\sum_{k=1}^m P_{ik}) - \ln \eta(i)) + \epsilon_\theta (\ln(\sum_{k=1}^n P_{kj}) - \ln \theta(j))$$

and

$$(\nabla_P^2 \Omega)_{ijkl} = \frac{\epsilon_P \delta_{ik}\delta_{jl}}{P_{ij}} + \frac{\epsilon_\eta \delta_{ik}}{\sum_{t=1}^m P_{it}} + \frac{\epsilon_\theta \delta_{jl}}{\sum_{t=1}^n P_{tj}}.$$

As we assume that $P_{ij} > 0$ for all $i, j$, all the terms above are well-defined. Besides, $\nabla_P \Omega$ is $C^1$ on $\eta, \theta$ and $\epsilon$. Therefore, we can show $P^\epsilon(C, \eta, \theta)$ is continuous not only on $\epsilon$ but also on $\eta$ and $\theta$ after checking Hessian. From implicit function theorem, if we show the above Hessian is invertible for $\epsilon_P > 0$, then the results of the proposition are true. Equivalently, it suffices to show that $\det H \neq 0$ where matrix $H$ is the flattened $\nabla_P^2 \Omega$ by mapping $(i, j, k, l) \mapsto (im + j, km + l)$.

**Invertibility of $H$.** Let $\mathbf{r}$ be the vector of reciprocals of row sums of $P$, i.e., $r_i = 1/\left(\sum_j P_{ij}\right)$, and similarly, let $\mathbf{c}$ be the vector of reciprocals of column sums of $P$, i.e., $c_j = 1/\left(\sum_i P_{ij}\right)$. Then

$$(\nabla_P^2 \Omega)_{ijkl} = \frac{\epsilon_P \delta_{ik}\delta_{jl}}{P_{ij}} + \epsilon_\eta \delta_{ik} r_i + \epsilon_\theta \delta_{jl} c_j.$$

Let $\phi$ be the map $(i, j) \mapsto (im + j)$, then $\phi$ induces a reshaping of $P$ to a vector of size $mn$, denoted by $P^\phi$. When there is no ambiguity, we may omit the $\phi$ superscript.

Further define $p^\phi$ as a vector of dimension $mn$ where $p_k^\phi = \epsilon_P / P_k^\phi$. By definition, $H^\phi = \epsilon_P(diag(p^\phi)) + \epsilon_\eta \mathbb{1}_m \otimes (diag(\mathbf{r})) + \epsilon_\theta(diag(\mathbf{c})) \otimes \mathbb{1}_n$ where $\mathbb{1}_k$ is the $k \times k$ matrix of ones, and $A \otimes B$ is Kronecker product (tensor product of matrices). Decompose $H = D + G$ where $D = \epsilon_P(diag(p^\phi))$ and $G = \epsilon_\eta \mathbb{1}_m \otimes (diag(\mathbf{r})) + \epsilon_\theta(diag(\mathbf{c})) \otimes \mathbb{1}_n$.

From now on, we may use $P$-row, $P$-column to represent $i, j$ style indices, and $G$-row, $G$-column or simply row/column to represent those of $G$, or the ones in range $[1, mn]$. $D$ is diagonal, and $\det G = 0$. Furthermore,

(∗) any row or column of $G$ with index $k$ can be represented by an entry position $(i, j)$ of $P$ by inverse of $\phi$, and any rows of indices $k_1, k_2, k_3, k_4$ corresponding to $(i_1, j_1), (i_1, j_2), (i_2, j_1), (i_2, j_2)$ (i.e., determined as intersections of two $P$-rows and two $P$-columns) is linearly dependent: $G_{(k_1, \_)} + G_{(k_4, \_)} - G_{(k_2, \_)} - G_{(k_3, \_)} = \mathbf{0}$, we denote this property as (∗).

Structure of $\det H$: Let $D = diag(p_1, p_2, \ldots, p_{mn})$, then $\det H$ is a polynomial on $p_k$'s with constant term 0. Each term in $\det H$ is of form $f(\mathcal{I}) \left(\prod_{k \notin \mathcal{I}} p_k\right)$ for each subset $\mathcal{I} \subseteq \{1, 2, \ldots, mn\}$, and the coefficient $f(\mathcal{I}) = \det G_{(\mathcal{I}, \mathcal{I})}$ where $G_{(\mathcal{I}, \mathcal{I})}$ is the submatrix with lines of indices not in $\mathcal{I}$, i.e., the entries of $G_{(\mathcal{I}, \mathcal{I})}$ are of the form $G_{ij}$ with $i \in \mathcal{I}$ and $j \in \mathcal{I}$.

Next we show that $f(\mathcal{I})$ is nonnegative for all $\mathcal{I}$, then with $p_k > 0$ for all $k$, we can conclude that $\det H > 0$. Since $\mathcal{I} \subseteq \{1, 2, \ldots, mn\}$, $\phi^{-1}(\mathcal{I}) \subseteq \{1, 2, \ldots, n\} \times \{1, 2, \ldots, m\}$, and $\phi$ is a bijection, we may not distinguish $\mathcal{I}$ from $\phi^{-1}(\mathcal{I})$, in order to make the statement neater.

**1.** [Operation-(∗) on $\mathcal{I}$]: We want to investigate the operations on $\mathcal{I}$ producing a subset $\mathcal{J}$ such that $f(\mathcal{I}) = f(\mathcal{J})$. By the properties of determinant, (∗) induces one operation: when $\mathcal{I}$ containing 4 integer pairs which can form the vertices of a rectangle, $f(\mathcal{I}) = 0$. Moreover, for any $k_1, k_2, k_3, k_4$ such indices in (∗), we can generate row $G_{(k_4, \_)}$ by $G_{(k_4, \_)} = G_{(k_2, \_)} + G_{(k_3, \_)} - G_{(k_1, \_)}$, then if $\{k_1, k_2, k_3\} \subseteq \mathcal{I}$, we can build $G_{(k_4, \_)}$ on any $G_{(k_i, \_)}$, thus the determinant $\det G_{(\mathcal{I}, \mathcal{I})}^{row} = \pm \det G_{(\mathcal{I}, \mathcal{I})}$ (positive for $k_2$ and $k_3$, negative for $k_1$). Similarly, if we follow the same operation on columns, we have $\det G_{(\mathcal{I}, \mathcal{I})}^{col} = \pm \det G_{(\mathcal{I}, \mathcal{I})}$. And when doing both, $\det G_{(\mathcal{I}, \mathcal{I})}^{col \cdot row} = \det G_{(\mathcal{I}, \mathcal{I})}$.

Therefore, we know that if $k_1, k_2, k_3 \in \mathcal{I}$, and $\mathcal{J} = \{k_4\} \cup \mathcal{I} \backslash \{k_i\}$ for any $i = 1, 2, 3$, then $f(\mathcal{I}) = f(\mathcal{J})$. Such operations changing $\mathcal{I}$ to $\mathcal{J}$ is denoted by operation-$*$. In short, an operation-$*$ moves an end of a small "L-shaped" set of 3 pairs along a $P$-row or a $P$-column, producing another L-shaped set of 3 pairs.

**2.** [Regularized form of $\mathcal{I}$, and decomposition of nondegenerate regularized form $\mathcal{I}^\sharp$ into L-shaped subsets]: Once $\mathcal{I}$ or any $\mathcal{J}$ equivalent to $\mathcal{I}$ via operations-$*$ contains 4 pairs satisfying condition ($*$), $f(\mathcal{I}) = 0$, then we call $\mathcal{I}$ degenerate. In decomposing $\mathcal{I}$, when we find it degenerate, we stop since $f(\mathcal{I})$ is known.

We decompose $\mathcal{I}$ as set of pairs inductively in the following way before stopping. Start with any $(i, j) \in \mathcal{I}$, we look for pairs of form $(i, l)$ and $(k, j)$ in $\mathcal{I}$, adding them into the subset $A_{(i,j)}$ containing $(i, j)$. Then check the degeneracy, by looking for whether $\mathcal{I}$ contains a point $(k, l)$ with $(i, l), (j, k) \in A_{(i,j)}$, whenever $\mathcal{I}$ is degenerate, we stop since $f(\mathcal{I}) = 0$. Next we enlarge $A_{(i,j)}$ by changing the set $\mathcal{I}$ to a regularized form using operation-$*$'s. For each $(k, l)$ with $(i, l) \in A_{(i,j)}$, then $(k, j)$ can be constructed on $(k, l)$ via an operation-$*$ with $(i, j)$ and $(i, l)$. Thus we modify $\mathcal{I}$ into $\mathcal{J} = (i, l) \cup \mathcal{I} \backslash (k, l)$ that $f(\mathcal{I}) = f(\mathcal{J})$, and adding $(i, l)$ into set $A_{(i,j)}$. Similar process can be done for those $(k, l) \in \mathcal{I}$ with $(k, j) \in A_{(i,j)}$.

After regularizing $\mathcal{I}$ and enlarging $A_{(i,j)}$ to maximum about $(i, j)$, we get a regularized form $\mathcal{J}$ of $\mathcal{I}$, with $f(\mathcal{I}) = f(\mathcal{J})$, and a component $A_{(i,j)}$ of L-shape. The set of $\mathcal{J} \backslash A_{(i,j)}$ has no elements of form $(k, l)$ with $(i, l) \in A_{(i,j)}$ or $(k, j) \in A_{(i,j)}$, as they are already moved to $A_{(i,j)}$ by operation-$*$. Therefore, $\mathcal{J} \backslash A_{(i,j)}$ is supported on a rectangular region by deleting all $P$-rows $(k, \_)$'s and $P$-columns $(\_, l)$'s where $k, l$'s occur in $A_{(i,j)}$.

Repeating the L-shaped component construction above for $\mathcal{J} \backslash A_{(i,j)}$, we can transform $\mathcal{I}$ into a regularized form (not unique or standard) $\mathcal{I}^\sharp$ and we have a decomposition $\mathcal{I}^\sharp = \bigcup A_{(i_t, j_t)}$ into L-shaped components, which do not intersect with each other. The name "regularized form" is given to the transformed set with a L-shaped decomposition, and since only operation-$*$ is applied, $f(\mathcal{I}) = f(\mathcal{I}^\sharp)$.

**3.** [Properties between the L-shaped subsets:] For each $\mathcal{I}$ which we did not conclude $f(\mathcal{I}) = 0$ in the last step, we get $\mathcal{I}^\sharp$ and a decomposition $\mathcal{I}^\sharp = \bigcup_{t \in T} A_t$ into L-shaped subsets.

The construction of components $A_t$ induces such a property: for two distinct components $A_t$ there is no elements $(i, j) \in A_t$ and $(k, l) \in A_s$, in normal words, the $A_t$ occupies certain $P$-rows and $P$-columns which is distinct from those of $A_s$.

For $(i, j)$ and $(k, l)$ with $i \neq k$ and $j \neq l$, $G_{im+j, km+l} = 0$ from the formula that $G_{im+j, km+l} = \epsilon_\eta r_i \delta_{ik} + \epsilon_\theta c_j \delta_{jl}$. Therefore, the decomposition $\mathcal{I}^\sharp = \bigcup_{t \in T} A_t$ induces a decomposition of matrix $G_{(\mathcal{I}^\sharp, \mathcal{I}^\sharp)}$ into blockwise diagonal matrix

$$
\begin{bmatrix}
G_{A_1, A_1} & 0 & \cdots & 0 \\
0 & G_{A_2, A_2} & \cdots & 0 \\
\vdots & & \ddots & \vdots \\
0 & 0 & \cdots & G_{A_t, A_t}
\end{bmatrix}
\tag{4}
$$

So for a decomposition $\mathcal{I}^\sharp = \bigcup_{t \in T} A_t$, we haves $f(\mathcal{I}^\sharp) = \prod_{t \in T} f(A_t)$

**4.** [$f(A)$ for an L-shaped component]: The last part is to show $f(A) > 0$ for all L-shaped components. Recall that $G_{im+j, km+l} = \epsilon_\eta r_i \delta_{ik} + \epsilon_\theta c_j \delta_{jl}$, so for $A$ an L-shaped component with $s$ $P$-rows and $t$ $P$-columns, $G_{(A,A)}$ in general is of form

$$
G_{(A,A)} =
\begin{bmatrix}
r_1 + c_1 & \cdots & r_1 & r_1 & 0 & \cdots & 0 \\
\vdots & \ddots & \vdots & \vdots & \vdots & \ddots & \vdots \\
r_1 & \cdots & r_1 + c_{t-1} & r_1 & 0 & \cdots & 0 \\
r_1 & \cdots & r_1 & r_1 + c_t & c_t & \cdots & c_t \\
0 & \cdots & 0 & c_t & c_t + r_2 & \cdots & c_t \\
\vdots & \ddots & \vdots & \vdots & \vdots & \ddots & \vdots \\
0 & \cdots & 0 & c_t & c_t & \cdots & c_t + r_s
\end{bmatrix}
\tag{5}
$$

Recall the formula $\det \begin{bmatrix} E & B \\ C & D \end{bmatrix} = \det(E)\det(D - CE^{-1}B)$ and the matrix determinant lemma

$$\det(diag(c) + r\mathbf{1}\mathbf{1}^T) = (1 + r\mathbf{1}^T diag(c)^{-1}\mathbf{1})\det(diag(c)) = \prod c_i(1 + \sum(r/c_i)).$$

If $s = 1$ or $t = 1$, the determinant of $G_{(A,A)}$ can be calculated directly by the matrix determinant lemma above.

If $s > 1$ and $t > 1$, we cut Eq. (5) into 4 blocks $\begin{bmatrix} E & B \\ C & D \end{bmatrix}$ where $E$ contains the upper left $t \times t$ part, $B$ is zero but the last row, $C$ is zero but the last column, $D$ is a matrix in a similar form as $E$.

According to the characters of $B, C$ stated above, it can be found that $CE^{-1}B = c_t^2 \mathbf{1} E_{t,t}^{-1} \mathbf{1}^T$ which is an $s \times s$-matrix. The entry $E_{t,t}^{-1} = \det E_{(1:t-1,1:t-1)}/\det E$ where $E_{(1:t-1,1:t-1)}$ is the matrix $E$ without the last row and last column, moreover, $E_{t,t}^{-1} = \left(\prod_1^{t-1} c_i(1 + \sum_1^{t-1}(r_1/c_i))\right) / \left(\prod_1^t c_i(1 + \sum_1^t(r_1/c_i))\right) = \dfrac{1 + \sum_1^{t-1}(r_1/c_i)}{c_t(1 + \sum_1^t(r_1/c_i))} < 1/c_t$. Therefore, $CE^{-1}B = \lambda \mathbf{1}\mathbf{1}^T$ with $\lambda < c_t$ and $D - CE^{-1}B = diag(r_{2:s}) + (c_t - \lambda)\mathbf{1}\mathbf{1}^T$, whose determinant is positive according to the matrix determinant lemma.

As a consequence, $\det G_{(A,A)} > 0$ for each L-shaped components $A$. So combining the discussions in [1-4], we have $\det H = \det(D + G) > 0$.

Then the implicit function theorem implies the differentiability of $P^\epsilon$ on $\epsilon$. $\qquad\square$

**Proposition 3.** *For any finite $s_P$, $s_\eta$, $s_\theta \geq 0$, the limit of $P^\epsilon$ exists as $\epsilon$ approaches to $(\infty, s_\eta, s_\theta)$. In fact, $\lim_{\epsilon \to (\infty, s_\eta, s_\theta)} P_{ij}^\epsilon = 1$ for all $i, j$ (Limit 1). Moreover, $P^\epsilon$ converges to the solution to*

$$\min\langle C, P\rangle - s_P H(P) + s_\theta KL(P^T\mathbf{1}|\theta), \text{ with constraint } P\mathbf{1} = \eta, \tag{6}$$

*as $\epsilon \to (s_P, \infty, s_\theta)$ (Limit 2). Similarly, $P^\epsilon$ converges to the solution to*

$$\min\langle C, P\rangle - s_P H(P) + s_\eta KL(P\mathbf{1}|\eta), \text{ with constraint } P^T\mathbf{1} = \theta, \tag{7}$$

*as $\epsilon \to (s_P, s_\eta, \infty)$ (Limit 3). And in the case when $\epsilon \to (s_P, \infty, \infty)$, the matrix $P^\epsilon$ converges to the EOT solution (Limit 4):*

$$\min\langle C, P\rangle - s_P H(P), \text{ with constraints } P^T\mathbf{1} = \theta \text{ and } P\mathbf{1} = \eta. \tag{8}$$

*When $\epsilon \to (\infty, \infty, s_\theta), (\infty, s_\eta, \infty)$ or $(\infty, \infty, \infty)$, the limit does not exist, but the directional limits can be calculated..*

*Proof.* Recall that $H(P) = -\sum_{ij}(P_{ij}\ln P_{ij} - P_{ij})$, $(\nabla_P H)_{ij} = -\ln P_{ij}$, and $H(P)$ is strictly concave, therefore $H$ has a unique maximum $mn$ at $P_{ij} = 1$, denoted by $\mathbb{1}$. Similarly, $KL(a|b) = \sum_i(a_i(\ln a_i - \ln b_i) - a_i + b_i)$, $\nabla_a KL(a|b)_i = \ln a_i - \ln b_i$, $KL$ is strictly convex, therefore $KL$ has a minimum 0 at $a_i = b_i$ for all $i$.

**Limit 1.** Shown by contradiction: When $\epsilon \to (\infty, s_\eta, s_\theta)$, suppose the limit $\lim_{\epsilon \to (\infty, s_\eta, s_\theta)} P_{ij}^\epsilon$ for some $(i, j)$ does not exist, or is not 1. Thus there is $e > 0$ that, for any $\delta > 0$ and $N > 0$, there exists a parameter $\epsilon_1 = (\epsilon_P, \epsilon_\eta, \epsilon_\theta)$ such that $\epsilon_P > N$, $|\epsilon_\eta - s_\eta| < \delta$ and $|\epsilon_\theta - s_\theta| < \delta$, satisfying $|P_{ij}^\epsilon - 1| > e$.

However, for any $0 < e < 1/2$, let $\delta = 1$, let $E = (1+e)\ln(1+e) - (1+e) + 1 > 0$, $\min \Omega(P; \epsilon) \leq \Omega(\mathbb{1}; \epsilon) < C$ for some $G > 0$ where $(\mathbb{1})_{ij} = 1$ for all $(i, j)$, and any $\epsilon \in \{(\epsilon_P, \epsilon_\eta, \epsilon_\theta) : s_\eta/2 < \epsilon_\eta < 3s_\eta/2, s_\theta/2 < \epsilon_\theta < 3s_\theta/2, \}$. So there is a $N > 0$ such that $NE > G + \max_{ij} C_{ij} + mn + L$ where $L = -\inf\{\epsilon_\eta KL(P\mathbf{1}|\eta) + \epsilon_\theta KL(P^t\mathbf{1}|\theta)\}$, meaning those $P$ with $|P_{ij} - 1| > e$ for some $(i, j)$ is not minimizing $\Omega$.

The contradiction indicates that $\lim_{\epsilon \to (\infty, s_\eta, s_\theta)} P_{ij}^\epsilon = 1$ for all $i, j$.

**Limit 2 & 3:** The situation of $\epsilon_\theta \to \infty$ and $\epsilon_\eta \to \infty$ are similar, so we only prove for $\epsilon_\theta \to \infty$ case. Let $\hat{P}$ denote the solution to Eq. (7).

Let $\hat{P}$ be the solution to the optimization with constraints. We first show that $\lim_{\epsilon \to (s_P, s_\eta, \infty)} \sum_{k=1}^{n} P_{kj}^{\epsilon} = \theta_j$.

This is similar to limit 1. Suppose the limit either does not exist or is not $\theta_j$, then there exists an $e > 0$ such that for any $N > 0, \delta > 0$, there exists $\epsilon_\theta > N$, $|\epsilon_\eta - s_\eta| < \delta$ and $|\epsilon_P - s_P| < \delta$, such that

$$\left| \sum_{k=1}^{n} P_{kj}^{\epsilon} - \theta_j \right| > e \tag{9}$$

for some $j$. Thus $KL((P^{\epsilon})^T \mathbf{1} | \theta) > E$ for some $E > 0$. Consider that $\langle C, P \rangle \geq 0, H(P) \geq -mn$ and $KL(P\mathbf{1}|\eta) \geq 0$ are lower bounded, we can take sufficiently large $N$ such that the $P^{\epsilon}$ satisfying Eq. (9) satisfy $\Omega(P^{\epsilon}; \epsilon) > \Omega(\hat{P}; \epsilon)$, making $P^{\epsilon}$ fail to optimize $\Omega(\cdot; \epsilon)$, which is a contradiction. Thus we have $\lim_{\epsilon \to (s_P, s_\eta, \infty)} \sum_{k=1}^{n} P_{kj}^{\epsilon} = \theta_j$.

For each $\epsilon = (\epsilon_P, \epsilon_\eta, \epsilon_\theta)$, let $\theta^{\epsilon}$ denote the $(P^{\epsilon})^T \mathbf{1}$, then for any $\epsilon$, the solution $P^{\epsilon}$ is also the solution to

$$\min_{P} \langle C, P \rangle + \epsilon_P H(P) + \epsilon_\eta KL(P\mathbf{1}|\eta), \text{ with constraint } P^T \mathbf{1} = \theta^{\epsilon}. \tag{10}$$

Denote $\Phi(P, \epsilon_P, \epsilon_\eta) := \langle C, P \rangle + \epsilon_P H(P) + \epsilon_\eta KL(P\mathbf{1}|\eta)$ When $\epsilon_P \in (0, \infty)$, the new objective function $\Phi(P, \epsilon_P, \epsilon_\eta)$ is continuous on $P$ and $\epsilon_P, \epsilon_\eta$, and each minimization problem gets a unique solution since the objective function is strictly convex. Therefore, the limit $\lim_{\epsilon \to (s_P, s_\eta, \infty)} P^{\epsilon} = \hat{P}$. We show this via contradiction:

Suppose the opposite, there exists some $\xi > 0$ such that $||P^{\epsilon} - \hat{P}||_2 > \xi$ for $\epsilon$ arbitrarily close to $(s_P, s_\eta, \infty)$. Let

$$\alpha := \inf_{P^T e = \theta, ||P - \hat{P}||_2 > \xi} \Phi(P, s_P, s_\eta) - \Phi(\hat{P}, s_P, s_\eta),$$

$\alpha > 0$ since the minimum $\hat{P}$ is unique and the objective is strictly convex. The sets $P^T e = \theta^{\epsilon}$ is compact since it is closed and bounded, so there exists bounds $b = (b_1, b_2, b_3)$ for $\epsilon = (\epsilon_P, \epsilon_\eta, \epsilon_\theta)$ such that in the bound where $|\epsilon_P - s_P| < b_1$, $|\epsilon_\eta - s_\eta| < b_2$ and $\epsilon_\theta > b_3$, $\max \Phi(P, s_P, s_\eta) - \Phi(P^\sharp, \epsilon_P, \epsilon_\eta) < \alpha/3$ for $P$ with $P^T e = \theta$ and $P^\sharp$ its Euclidean projection to $\{P^T e = \theta^{\epsilon}\}$, and $\max \Phi(P, \epsilon_P, \epsilon_\eta) - \Phi(P^\flat, s_P, s_\eta) < \alpha/3$ for $P$ with $P^T e = \theta^{\epsilon}$ and $P^\flat$ its Euclidean projection to $\{P^T e = \theta\}$.

Let $\epsilon$ be a parameter in the above bound $b$ to $(s_P, s_\eta, \infty)$, where $P = argmin_{P^T e = \theta^{\epsilon}} \Phi(P, \epsilon_P, \epsilon_\eta)$ is $\xi$ far from $\hat{P}$. Then $\Phi(P, \epsilon_P, \epsilon_\eta) > \Phi(P^\flat, s_P, s_\eta) - \alpha/3 > \Phi(\hat{P}, s_P, s_\eta) + 2/3\alpha > \Phi(\hat{P}^\sharp, \epsilon_P, \epsilon_\eta) + \alpha/3 > \Phi(\hat{P}^\sharp, \epsilon_P, \epsilon_\eta)$, which is a contradiction to the assumption that $P$ is the argmin.

**Limit 4:** Similar to the previous two limits, we can say that $\lim_{\epsilon \to (s_P, \infty, \infty)} \sum_{k=1}^{n} P_{kj}^{\epsilon} = \theta_j$ and $\lim_{\epsilon \to (s_P, \infty, \infty)} \sum_{k=1}^{m} P_{ik}^{\epsilon} = \eta_i$. Then the problem becomes the EOT problem, which has a unique solution.

**Boundaries at $\epsilon_\eta = 0$ or $\epsilon_\theta = 0$:** It is simple to check the continuity when $\epsilon_\eta \to 0$ or $\epsilon_\theta \to 0$. From Prop. 2, the continuity and differentiability hold for $\epsilon_\eta \to 0$ or $\epsilon_\theta \to 0$ when $\epsilon_P > 0$.

**Nonexistence of the limits when $\epsilon_P, \epsilon_\eta \to \infty$, and directional limits:** Let a sequence $\epsilon_1, \epsilon_2, \ldots$ where $\epsilon_i = (\epsilon_P^i, \epsilon_\eta^i, \epsilon_\theta^i)$ satisfy $\lim \epsilon_P^i = \lim \epsilon_\eta^i = \infty$ and $\lim(\epsilon_\eta^i / \epsilon_P^i) = t$, then the limit $P$ of $P^{\epsilon}$ satisfy $P_{ij} = t(\ln c_j - \ln n)/(t + 1)$, since the limit minimizes the following objective function

$$H(P) + tKL(P\mathbf{1}|\eta).$$

The reason is, as $\sum \eta_i = 1$, $H(P)$ and $KL(P\mathbf{1}|\eta)$ cannot vanish for the same $P$, thus the minima of objective function approaches to infinity, therefore the finite terms $\langle C, P \rangle$ and $\epsilon_\theta KL(P^T \mathbf{1}|\theta)$ tend to have no effect on the minimal point $P$ as $\epsilon_P, \epsilon_\eta$ increases to infinity.

A direct consequence of the above discussion is, when $t$ changes, the limits $P$ of those sequences changes, which indicates that the limit of $P^{\epsilon}$ as $\epsilon \to (\infty, \infty, s_\theta)$ fails to exist. And similar situation happens when $\epsilon \to (\infty, s_\eta, \infty)$

**Nonexistence of the limits when** $\epsilon_P, \epsilon_\eta, \epsilon_\theta \to \infty$**, and directional limits :** Similar to the discussions above, let the sequence $\epsilon_1, \epsilon_2, \ldots$ where $\epsilon_i = (\epsilon_P^i, \epsilon_\eta^i, \epsilon_\theta^i)$ satisfy $\lim_{i \to \infty} \epsilon_i = (\infty, \infty, \infty)$. Further let $\lim(\epsilon_\eta^i/\epsilon_P^i) = u$, $\lim(\epsilon_\theta^i/\epsilon_P^i) = w$, then $P^{\epsilon_i}$ converges to the solution to the problem

$$H(P) + uKL(P1|\eta) + wKL(P^T 1|\theta),$$

which could be considered as another UOT problem with cost function constantly 0.

$\square$

**Corollary 4.** *Consider a UOT problem with cost* $C = -\log \mathbb{P}(d|h)$*, marginals* $\theta = \mathbb{P}(h)$*,* $\eta \in \mathcal{P}(\mathcal{D})$*. The optimal UOT plan* $P^{(1, \epsilon_\eta, \epsilon_\theta)}$ *converges to the posterior* $\mathbb{P}(h|d)$ *as* $\epsilon_\eta \to 0$ *and* $\epsilon_\theta \to \infty$*. Bayesian inference is a special case of GBT with* $\epsilon = (1, 0, \infty)$*.*

*Proof.* As direct application of Limit 3 of Proposition 3, we only need to show that the optimal plan $P^{(1,0,\infty)}$ is propositional to the posterior $P(h|d)$.

$$P^{(1,0,\infty)} = \underset{P \in U(\theta)}{\arg\min} \; K(P) := \underset{P \in U(\theta)}{\arg\min} \; \{\langle C, P \rangle - H(P)\}. \tag{11}$$

where $U(\theta) = \{P \in \mathcal{M}(D \times H)|P^T \mathbf{1} = \theta\}$.

Let $\boldsymbol{\lambda} \in \mathbb{R}^{+m}$, consider the corresponding Lagrangian problem:

$$L(P, \boldsymbol{\lambda}) := \langle C, P \rangle - H(P) + \langle \boldsymbol{\lambda}, (P^T \mathbf{1} - \theta) \rangle$$

Partial derivatives $\partial_{P_{ij}} = 0$ and $\partial_{\lambda_j} L = 0$ result the following system of equations:

$$\log P_{ij} - \log P(d_i|h_j) + \lambda_j = 0 \quad \sum_i P_{ij} - P(h_j) = 0 \tag{12}$$

Calculation shows that the solution to Equation 12 is $P_{ij} = \frac{P(d_i|h_j)P(h_j)}{\sum_i P(d_i|h_j)} = P(d_i|h_j)P(h_j) \propto P(h_j|d_i)$. Hence the proof is completed. $\square$

**Corollary 5.** *Consider a UOT problem with* $\theta \in \mathcal{P}(\mathcal{H})$*,* $\eta = \mathbb{P}(d)$*. The optimal UOT plan* $P^{(\epsilon_P, \infty, 0)}$ *converges to* $\eta \otimes \mathbf{1}$ *as* $\epsilon_P \to \infty$*. Frequentist Inference is a special case of GBT with* $\epsilon = (\infty, \infty, 0)$*.*

*Proof.* As direct application of Proposition 3, we only need to show that $P^{(\infty,\infty,0)} = \eta \otimes \mathbf{1}$. Notice that Eq 1 is equivalent to

$$P^{(\infty,\infty,0)} = \underset{P \in (\mathbb{R}_{\geq 0})^{n \times m}}{\arg\min} \; H(P), \text{ with constraint } P\mathbf{1} = \eta \tag{13}$$

Hence $P^{(\infty,\infty,0)} = \eta \otimes \mathbf{1}$. $\square$

**Corollary 6.** *Let cost* $C = -\log M$*, marginals* $\theta = \mathbb{P}(h)$ *and* $\eta = \mathbb{P}(d)$*. The optimal UOT plan* $P^{(1,\epsilon_\eta,\epsilon_\theta)}$ *converges to the optimal plan* $L$ *as* $\epsilon_\eta \to \infty$ *and* $\epsilon_\theta \to \infty$*. Cooperative Inference is a special case of GBT with* $\epsilon = (1, \infty, \infty)$*, which is exactly entropic Optimal Transport (Cuturi, 2013).*

*Proof.* According to proposition 17, $L = P^{(1,\infty,\infty)}$, and the convergence result is a direct application of Limit 4 of Proposition 3 $\square$

**Corollary 7.** *Consider a UOT problem with cost* $C = -\log \mathbb{P}(d, h)$*,* $m = n$*, and marginals* $\theta = \eta$ *are uniform. The optimal UOT plan* $P^{(\epsilon_P, \epsilon_\eta, \epsilon_\theta)}$ *approaches to a diagonal matrix as* $\epsilon_\eta, \epsilon_\theta \to \infty$ *and* $\epsilon_P \to 0$*. In particular, discriminative learner is a special case of GBT with* $\epsilon = (0, \infty, \infty)$*, which is exactly classical Optimal Transport (Villani, 2008).*

*Proof.* Limit 4 of Proposition 3 implies the convergence of $P^{(\epsilon_P, \epsilon_\eta, \epsilon_\theta)} \to P^{(0, \infty, \infty)}$ as $\epsilon_\eta, \epsilon_\theta \to \infty$ and $\epsilon_P \to 0$. When $m = n$, $P^{(0, \infty, \infty)}$ is a permutation matrix is the result of Wang et al. (2020b)[Proposition 8].

$\square$

**Proposition 8.** *In GBT with $\epsilon_\theta = \infty$, cost $C$ and current belief $\theta$. The learner updates $\theta$ with UOT plan in the same way as applying Bayes rule with likelihood from $P^\epsilon(C, \eta, \theta)$, and prior $\theta$.*

*Proof.* From Algorithm 1, for a general data point $d^i$ chosen, the GBT takes the vector normalization of some row $P^\epsilon$, i.e., $\theta' = P^\epsilon_{(i,\_)} / (\sum_j P^\epsilon_{ij})$.

On the other hand, when we apply Bayes rule to $P^\epsilon$, prior is $\theta = \mathbb{P}(h)$, likelihood $\mathbb{P}(d|h)$ is the column normalization of $P^\epsilon$, satisfying $\mathbb{P}(d^i|h^j) = P^\epsilon_{ij} / (\sum_i P^\epsilon_{ij}) = P^\epsilon_{ij} / \theta_j$. The last equality is because $\theta(i) = \sum_j P^\epsilon_{ij}$ when $\epsilon_\theta = \infty$. So the posterior $\mathbb{P}(h|d^i)$ is the vector normalization of $\mathbb{P}(d^i|h)\mathbb{P}(h)$, by $\mathbb{P}(d^i|h^j)\mathbb{P}(h^j) = P^\epsilon_{ij}/\theta_j * \theta_j = P^\epsilon_{ij}$. Therefore, $\mathbb{P}(h^j|d^i) = \theta'(h^j)$. $\square$

Now, we introduce some notations will be used in the following proofs.

**Notations.** Denote the set of all possible belief by $\Delta = \mathcal{P}(\mathcal{H})$. Distribution of $\Theta_k$ is denoted by $\mu_k$. We only consider the case where no two hypotheses are the same in $\mathcal{H}$. Hence we make the following assumption that columns of $\exp(-\epsilon_P C)$ are not differ by a multiplicative scalar, i.e. columns of $C$ are not differ by an additive scalar.

**Lemma 10.** *For $\epsilon = (\epsilon_P, \infty, \infty)$, $\epsilon_P \in (0, \infty)$, given cost $C$ with initial belief $\theta_0 \in \mathcal{P}(\mathcal{H})$ and fixed teaching and learning distribution $\eta_k = \eta \in \mathcal{P}(\mathcal{D})$ for all $k$, then the belief random variables $(\Theta_k)_{k \in \mathbb{N}}$ have the same expectation on $h$: $\mathbb{E}_{\Theta_k}[\theta(h)] = \theta_0(h)$.*

*Proof.* We start the proof by showing $\mathbb{E}_{\Theta_k}[\theta(h)] = \mathbb{E}_{\Theta_{k-1}}[\theta(h)]$ for $k \geq 1$. Notice that given cost $C$ and data marginal $\eta$, an observed data $d \in \mathcal{D}$ and UOT planning uniquely determines a map from a learner's initial belief $\theta_{k-1}$ to one's posterior belief $\theta_k$. Denote this map by $T_d : \theta_{k-1} \mapsto \theta_k$. Let the distribution of $\Theta_{k-1}$ over $\mathcal{P}(\mathcal{H})$ be $\mu_{k-1}$, denote its support by $S_{k-1}$. Then the following holds:

$$\mathbb{E}_{\Theta_k}[\theta(h^j)] = \sum_{\theta \in S_{k-1}} \mu_{k-1}(\theta) \sum_{d^i \in \mathcal{D}} \eta^i T_{d^i}(\theta)(h^j) = \sum_{\theta \in S_{k-1}} \mu_{k-1}(\theta) \sum_{d^i \in \mathcal{D}} \eta^i \frac{M_k(i,j)}{\eta^i}$$

$$= \sum_{\theta \in S_{k-1}} \mu_{k-1}(\theta) \sum_{d^i \in \mathcal{D}} M_k(i,j) = \sum_{\theta \in S_{k-1}} \mu_{k-1}(\theta)\theta(h^j) = \mathbb{E}_{\Theta_{k-1}}[\theta(h)]$$

Hence $\mathbb{E}_{\Theta_k}[\theta(h)] = \mathbb{E}_{\Theta_{k-1}}[\theta(h)] = \cdots = \mathbb{E}_{\Theta_0}[\theta(h)] = \theta_0(h)$.

$\square$

**Theorem 11.** *Consider a learning problem with initial belief $\theta_0 \in \mathcal{P}(\mathcal{H})$, and the true hypothesis $h^*$ defined by $\eta \in \mathcal{P}(\mathcal{D})$. If the learner's data distribution $\eta_k = \eta$, then belief random variables $(\Theta_k)_{k \in \mathbb{N}}$ converge to the random variable $Y$ in probability, where $Y = \sum_{h \in \mathcal{H}} \theta_0(h)\delta_h$ and $Y$ is supported on $\{\delta_h\}_{h \in \mathcal{H}}$ with $P(Y = \delta_h) = \theta_0(h)$ for $\epsilon_\eta = \epsilon_\theta = \infty$ and $\epsilon_P \in (0, \infty)$.*

*Proof.* Step 1: First, we show the following claim inspired the proof proposition 5.1 in Wang et al. (2020a)

**Claim**: $\lim_{k \to \infty} \mu_k(\Delta_\epsilon) = 0$, for any $\epsilon > 0$, where $\Delta_\epsilon := \{\theta \in \Delta : \theta(h) \leq 1 - \epsilon, \forall h \in \mathcal{H}\}$.

Assume the claim does not hold, then there exists $\alpha > 0$ and a subsequence $(\mu_{k_i})_{i \in \mathbb{N}}$ such that $\mu_{k_i}(\Delta_\epsilon) > \alpha$ for all $i$.

Let the center of $\Delta$ be $u$, we define $L(\mu) := \mathbb{E}_\mu f(\theta)$, where $f(\theta) = \|\theta - u\|_2^2$, ($f$ may also be chosen as entropy $H(\theta)$). Then $L(\mu_{k+1}) = \mathbb{E}_{\mu_k}(\mathbb{E}_{d \sim \eta} f(T_d(\theta)))$.

Notice that $f$ is strictly convex, by Jensen's inequality,

$$\mathbb{E}_{d\sim\eta}f(T_d(\theta)) \overset{(a)}{\geq} f(\mathbb{E}_{d\sim\eta}T_d(\theta)) \overset{(b)}{=} f(\theta) \tag{14}$$

Here $(b)$ holds because:

$$\mathbb{E}_{d\sim\eta}T_d(\theta) \overset{(c)}{=} \sum_{d^i\in\mathcal{D}} \eta^i \cdot (M_k(i,\_)\backslash\eta^i) = \sum_{d^i\in\mathcal{D}} M_k(i,\_) \overset{(d)}{=} \theta \tag{15}$$

$(c), (d)$ hold since $M_k$ has marginals $\eta, \theta$.

Moreover, equality holds in $(a)$ if and only if $T_d(\theta) = \theta$ for all $d \in \mathcal{D}$. Thus rows of $M_k$ are the same up to a scalar. This implies either (1) only one column of $M_k$ is none zero, thus $\Theta_k \equiv \delta_h$ for some $h$ or (2) $M_k$ has at least two columns are differed by a scalar.

In the case of (1), if $\theta_0 \neq \delta_h$, $\Theta_k \equiv \delta_h$ is contradict to Lemma 10. Otherwise, $Y = \delta_h$, the result holds. In the case of (2), according to Wang et al. (2019), $M_k$ is cross-ratio equivalent to $\exp(-\epsilon_P C)$, hence $\exp(-\epsilon_P C)$ has two columns differ by a multiplicative scalar, contradict to the assumption.

Thus for any $\theta \in \Delta_\epsilon$, $\mathbb{E}_{d\sim\eta}f(T_d(\theta)) > f(\theta)$. Therefore $L(\mu_{k+1}) > L(\mu_k)$ for any $k$.

Moreover, notice that $\Delta_\epsilon$ is compact, there is a lower bound $\beta > 0$, such that $\mathbb{E}_{d\sim\eta}f(T_d(\theta)) - f(\theta) > \beta$ for all $\theta \in \Delta_\epsilon$. Therefore:

$$\begin{aligned} L(\mu_{k_i+1}) &= \mathbb{E}_{\theta_{k_i+1}\in\Delta_\epsilon}(\mathbb{E}_{d\sim\eta}f(T_d(\theta))) + \mathbb{E}_{\theta_{k_i+1}\in\Delta\backslash\Delta_\epsilon}(\mathbb{E}_{d\sim\eta}f(T_d(\theta))) \\ &> \mathbb{E}_{\theta_{k_i}\in\Delta_\epsilon}(f(\theta)) + \mathbb{E}_{\theta_{k_i}\in\Delta\backslash\Delta_\epsilon}(f(\theta)) + \alpha*\beta \\ &= L(\mu_{k_i}) + \alpha*\beta. \end{aligned} \tag{16}$$

Thus $L(\mu_{k_i+s}) > L(\mu_{k_i}) + s*\alpha*\beta \to \infty$ as $s \to \infty$. On the other hand, by definition, $f(\theta)$ is bounded above by the diameter of $\Delta$ under $l^2$ norm, so $L(\mu)$ is also bounded above. Contradiction! Therefore, the Claim holds.

Step 2. We show $\lim_{k\to\infty} P(\Theta_k \in \Delta^h_{1-\epsilon}) = \lim_{k\to\infty} \mu_k(\Delta^h_{1-\epsilon}) = \theta_0(h)$, for all $h \in \mathcal{H}$ where $\Delta^h_{1-\epsilon} := \{\theta \in \Delta : \theta(h) > 1 - \epsilon\}$.

For a fixed $h \in \mathcal{H}$, we have:

$$\begin{aligned} \theta_0(h) &\overset{(a)}{=} \mathbb{E}_{\Theta_k}(\theta(h)) \overset{(b)}{=} \mathbb{E}_{\theta_k\in\Delta^h_{1-\epsilon}}(\theta(h^j)) + \mathbb{E}_{\theta_k\in\Delta^u_{1-\epsilon}}(\theta(h)) + \mathbb{E}_{\theta_k\in\Delta_\epsilon}(\theta(h)) \\ &\overset{(c)}{\leq} \mu_k(\Delta^h_{1-\epsilon}) \cdot 1 + \mu_k(\Delta^u_{1-\epsilon}) \cdot \epsilon + \mu_k(\Delta_\epsilon) \cdot 1 \\ &= \mu_k(\Delta^h_{1-\epsilon}) + \epsilon + \mu_k(\Delta_\epsilon) \end{aligned}$$

where $\Delta^u_{1-\epsilon}$ denotes the union of all the other corners of $\Delta$, i.e. $\Delta^u_{1-\epsilon} := \cup_{h'\in\mathcal{H}\backslash h}\Delta^{h'}_{1-\epsilon}$. Here $(a)$ is direct application of Lemma 10; $(b)$ holds since $\Delta = \Delta^h_{1-\epsilon} \cup \Delta^u_{1-\epsilon} \cup \Delta_\epsilon$. $(c)$ holds because in general $\theta(h^j) < 1$, and $\theta(h^j) < \epsilon$ for any $\theta \in \Delta^u_{1-\epsilon}$. Therefore, $0 \leq \theta_0(h) - \mu_k(\Delta^h_{1-\epsilon}) \leq \epsilon + \mu_k(\Delta_\epsilon) \to \epsilon$ as $k \to \infty$ hold for any choice of $\epsilon$. Pick a sequence of $\epsilon \to 0$, we have that $\lim_{k\to\infty}\mu_k(\Delta^h_{1-\epsilon}) = \theta_0(h)$.

Hence combining results from Step 1 and Step 2, we have shown $\Theta_k$ converges to $Y$ in probability: $P(|\Theta_k - Y| > \epsilon) \leq \mu_k(\Delta_\epsilon) + \sum_{h\in\mathcal{H}}(\theta_0(h) - \mu_k(\Delta^h_{1-\epsilon})) \to 0$ as $k \to \infty$. Hence the proof is completed. $\qquad\square$

**Corollary 12.** *Given a fixed data sequence $d_i$ sampled from $\eta$, if $\theta_k$ converges to $\delta_{h^j}$, then the $j$-th column of $M_k$ converges to $\eta$.*

*Proof.* For $\epsilon > 0$, there exists $N > 0$ such that $\theta_k(h^j) > 1-\epsilon$ for any $k > N$. So $\sum_{j'\neq j} M_k(i,j') < \epsilon$ for any $d_i \in \mathcal{D}$, on the other hand $\sum_{j'} M_k(i,j') = \eta_i$. This implies that $\eta_i - \epsilon < M_k(i,j) < \eta_i$, so $M_k(i,j) \to \eta_i$ as $\epsilon \to 0$. Therefore the $j$-th column of $M_k$ converges to $\eta$. $\qquad\square$

**Proposition 13.** *Consider a learning problem with cost $C$, initial belief $\theta_0 \in \mathcal{P}(\mathcal{H})$, the true hypothesis $h^*$ defined by $\eta \in \mathcal{P}(\mathcal{D})$. If the learner updates the estimation $\eta_k$ with observed data (sampled from $\eta$) as stated above, then belief random variables $(\Theta_k)_{k \in \mathbb{N}}$ satisfies that for any $s > 0$, $\lim_{k \to \infty} \sum_{h \in \mathcal{H}} P(\Theta(h) > 1 - s) = 1$. As a consequence, $M_k$ as the transport plan has a dominant column $(h^j)$ with total weights $> 1 - s$, and $|(M_k)_{ij} - \eta_k(i)| < s$. In fact, as long as the sequence of $\eta_k$ as random variables converges to $\eta$ in probability, the above proposition holds.*

*Proof.* The proof is similar to Step 1 of Theorem 11. The major difference is that data are sampled from $\eta$ in each step, whereas the learner only has an estimation $\eta_k$ at round $k$. Therefore, under current condition, equality $(b)$ of Eq 14 need to be modified as following:

$$\mathbb{E}_{d \sim \eta} T_d(\theta_k) = \sum_{d^i \in \mathcal{D}} \eta^i \cdot (M_k(i, \_) \backslash \eta_k^i) = \sum_{d^i \in \mathcal{D}} M_k(i, \_) \cdot \frac{\eta^i}{\eta_k^i} = \theta_k \odot \mathbf{v_k}. \tag{17}$$

where $\mathbf{v_k} = (\frac{\eta^i}{\eta_k^i})$ is a vector of the size of the data set $\mathcal{D}$, and $\odot$ represents element-wise product. Hence $\mathbb{E}_{d \sim \eta} f(T_d(\theta_k)) = f(\theta_k \odot \mathbf{v_k})$ holds for all $\theta_k \in \Delta$. Since $\eta_k \to \eta$ as $k \to \infty$. For any $\alpha * \beta > 0$, there exists $N > 0$ such that for $k > N$, $|1 - \frac{\eta^i}{\eta_k^i}| < \sqrt{\frac{\alpha * \beta}{2n}}$. Hence: $|f(\theta_k \odot \mathbf{v_k}) - f(\theta_k)| \le \frac{\alpha * \beta}{2}$. Then corresponding to Eq 16, for $k_i > N$, we have:

$$\begin{aligned}
L(\mu_{k_i + 1}) &= \mathbb{E}_{\theta_{k_i + 1} \in \Delta_\epsilon}(\mathbb{E}_{d \sim \eta} f(T_d(\theta))) + \mathbb{E}_{\theta_{k_i + 1} \in \Delta \backslash \Delta_\epsilon}(\mathbb{E}_{d \sim \eta} f(T_d(\theta))) \\
&> \mathbb{E}_{\theta_{k_i} \in \Delta_\epsilon}(f(\theta_k \odot \mathbf{v_k})) + \mathbb{E}_{\theta_{k_i} \in \Delta \backslash \Delta_\epsilon}(f(\theta_k \odot \mathbf{v_k})) + \alpha * \beta \\
&> \mathbb{E}_{\theta_{k_i} \in \Delta_\epsilon}(f(\theta_k)) + \mathbb{E}_{\theta_{k_i} \in \Delta \backslash \Delta_\epsilon}(f(\theta_k)) - \frac{\alpha * \beta}{2} + \alpha * \beta \\
&= L(\mu_{k_i}) + \frac{\alpha * \beta}{2}.
\end{aligned}$$

Hence the contradiction on the upper bound of $L(\mu_{k_i+1})$ still holds, which shows the claim that: $\lim_{k \to \infty} \mu_k(\Delta_\epsilon) = 0$. So $\lim_{k \to \infty} \sum_{h \in \mathcal{H}} P(\Theta(h) > 1 - s) = 1$. The proof for the second part of the proposition follows exactly as Corollary 12. $\square$

**Proposition 15.** *For $\epsilon = (\epsilon_P, \epsilon_\eta, 0)$ with $\epsilon_P \in (0, \infty)$, as $\eta_k \to \eta$ almost surely, the sequence $\Theta_k$ of posteriors as a sequence of random variables converges in probability to variable $\Theta$, where $\mathbb{P}(\Theta = \mathbf{v}^i) = \eta(i)$ and $\mathbf{v}^i = P_{(i, \_)} / \left( \sum_{j=1}^m P_{ij} \right)$ and $P = P^\epsilon(C, \eta, \theta)$. Therefore, for any $s > 0$, $\lim_{k \to \infty} \sum_{h \in \mathcal{H}} \mathbb{P}(|\Theta_k(h) - 1| < s) = 0$ for generic (for all but in a closed subset) cost $C$ and $\eta$, $\theta$.*

*Proof.* First, $\epsilon_\theta = 0$ means that $P^\epsilon(C, \eta, \theta)$ is independent of $\theta$. Therefore, $M_k = P^\epsilon(C, \eta_k, \theta)$ and has a limit $P^\epsilon(C, \eta, \theta)$, regardless of the concrete posterior $\theta_k$. From construction of GBT, the posterior $\Theta_k$ is determined by $\mathbb{P}(\Theta_k = \mathbf{w}_k^i) = \eta(i)$ where $\mathbf{w}_k^i = (M_k)_{(i, \_)} / \sum_{j=1}^m (M_k)_{ij}$. Given the coupling $(\Theta_k, \Theta)$ by setting only $\mathbb{P}(\Theta_k = \mathbf{w}_k^i, \Theta = \mathbf{v}^i) = \eta(i)$ for each $i$, we may calculate $\mathbb{P}(|\Theta_k - \Theta| < s)$ converge to 1 as $M_k$ converge to $P^\epsilon(C, \eta, \theta)$.

For generic $C, \eta, \theta$, the probability of $P^\epsilon(C, \eta, \theta)$ having a row with only one nonzero entry is 0.

$\square$

**Remark:** As $\eta_k \to \eta$ almost surely, for any $e > 0$, there exists $N > 0$, such that, when $k > N$, the probability of having $\eta_k$ $e$-close to $\eta$ is 1. Thus in almost all episodes, with generic $C, \eta, \theta$, when $e$ is small enough, for any $||\eta' - \eta|| < e$ (using $p - \infty$ norm, same for below), the row-normalized (to $\mathbb{1}_n$) UOT plans

$$\max_i ||P_r^\epsilon(C, \eta', \theta)_{(i, \_)} - P_r^\epsilon(C, \eta', \theta)_{(i, \_)}|| < \frac{1}{4} \min_{i,j} ||P_r^\epsilon(C, \eta, \theta)_{(i, \_)} - P_r^\epsilon(C, \eta, \theta)_{(j, \_)}||$$

where $P_r^\epsilon$ is the row normalization of $P^\epsilon$.

Therefore, for such $e$, we may find an $N > 0$ such that for any $k, k' > N$, $P_r^\epsilon(C, \eta_k, \theta) \neq P_r^\epsilon(C, \eta'_k, \theta)$. However, for generic $\eta$, say, no entry of $\eta$ is 0, $||\theta_k - \theta'_k|| <$ when $k, k' > N$ and $d_k \neq d_{k'}$. Thus the posterior sequence of almost every episode fails to converge.

## C  ADDITIONAL SIMULATIONS

Interpolation between learning models can be investigated properly under GBT. Human learners appear to be capable of moving between different learning models gradually. Consider an individual at a carnival who is playing a game. At each of 10 trials, a bit of information is provided, but the available reward decreases. The individual has a pool of tickets with which they can bet on the outcome at each trial. The question is how the individual should update their beliefs in order to maximize their rewards. On the first trial, their belief update, in order to accurately reflect the evidence, should follow Bayes rule. However, for the last trial, one should focus bets on the most probable outcome in order to maximize chances for rewards, that is, their beliefs should be optimized for discriminating among the possible outcomes. GBT offers a coherent way of interpolating between these two approaches to provide candidate strategies on the intermediate steps. Such situations are common where there is an explicit constraint on the time horizon after which point no further evidence can be obtained, and there are incentives to act early, rather than to wait until evidence has fully accumulated; for example, identifying dangerous situations (tiger or not? poisonous or not?).

We now demonstrate how continuity of GBT (section 3.1) allows one to gradually interpolate between Bayesian and discriminative learning over steps (rather than a sharp switch).

### C.1  SIMULATION SETUP

Suppose a learner who observes data sampled from a true hypothesis $\mathbb{P}(d|h^*)$, and needs to make a conclusion on whether $h^*$ is one of the hypotheses in $\mathcal{H}$ within a fixed number $N$ of observations.

Here we compare a baseline learner who utilizes Bayesian inference ($\epsilon = (1, 0, \infty)$) on the first $N - 1$ observations, and switch to discriminative learning ($\epsilon = (0, \infty, \infty)$) on the last observation, against learners who interpolate from Bayesian to discriminative learning gradually along a sequence of models on curves in GBT. Two curves along with intermediate models are shown red and orange in Figure 5.

We take a random sampled $M$ of shape $4 \times 4$ as an example,

$$M = \begin{bmatrix} 0.225779 & 0.014886 & 0.433787 & 0.050735 \\ 0.613779 & 0.322347 & 0.172658 & 0.109262 \\ 0.069799 & 0.620178 & 0.29083 & 0.243635 \\ 0.090643 & 0.042588 & 0.102725 & 0.596368 \end{bmatrix}.$$

Thus $|\mathcal{H}| = |\mathcal{D}| = 4$. Set $N = 10$ and start from uniform $\theta = (0.25, 0.25, 0.25, 0.25)$.

Simulation details: We perform 40000 trials in total. For each trial $s$ (or say each episode), we uniformly sample $X_s \in \mathcal{P}(\mathcal{H})$, and let the true hypothesis $h^*$ be the covex combination of elements in $\mathcal{H}$ with coefficients given by $X_s$. While teaching the episode, in each round, we sample a hypothesis $h \in \mathcal{H}$ following $X_s$, then sample a data $d$ following the column of $M$ corresponding to $d$. During inference, we set $\eta_k$ by counting the frequency of each $d \in \mathcal{D}$ (starting from 1 to avoid 0 in $\eta_k$) and then normalize, as stated in (**RS**) model in Sec. 4.1.

### C.2  RESULTS

Following paths shown in Fig. 5, for baseline (blue, left), path 1 (orange, middle), and path 2 (red, right), the distribution of maximal component of each posterior at round 10 are shown in histograms of 30, and the entropy of these posteriors are plotted in the lower three figures.

In the upper figures, comparing to the baseline (blue), weights are concentrated more on the right bars for the gradual interpolations (orange and red). Thus learning tends to be more conclusive along these paths. Here conclusiveness means that the ability of getting a conclusion (one component of the posterior eventually becoming dominant). Furthermore, the entropy distributions shown in the lower figures also illustrate this point, as compare to baseline, gradual interpolations have lower entropy.

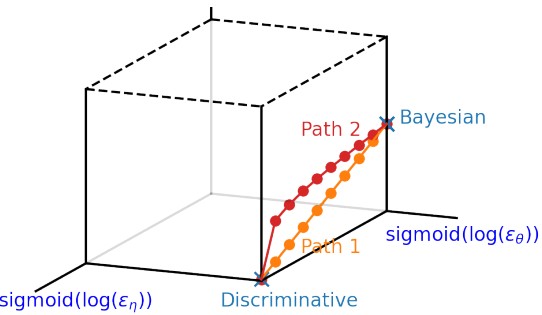

Figure 5: Baseline (sharp change) and two paths we follow on the parameter space of GBT.

Numerical results: entropy of baseline: mean 0.1888, standard deviation 0.2858, entropy along path 1: mean 0.0097, standard deviation 0.0686 entropy along path 2: mean 0.0571, standard deviation 0.1584.

It is necessary to consider that, the two paths and interpolations are chosen for demonstration purpose, by no means they are optimal. However, we believe GBT is capable of facilitating exploration of such optimization.

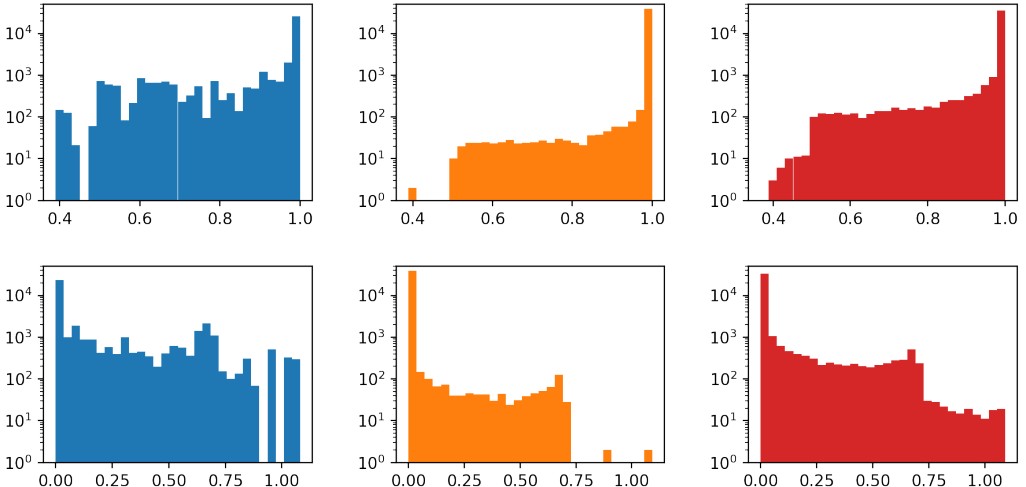

Figure 6: Results. Upper: distribution of maximal component of posterior. Lower: Entropy distribution of posteriors. Left: baseline. Middle: along path 1. Right, along path 2.

