# OpenReview forum: "Generalized Belief Transport"
_ICLR.cc/2023/Conference — Submitted to ICLR 2023_

### Official Review · Reviewer_KwJp · 2022-10-21

**Confidence:** 3
**Correctness:** 3
**Technical Novelty And Significance:** 3
**Empirical Novelty And Significance:** Not applicable
**Recommendation:** 5

**Clarity, Quality, Novelty And Reproducibility:**

First of all, a detailed related work section should be included just after the introduction. For the moment, we just have a short paragraph (Section 5) that briefly mentions existing studies without relating them to the present framework. Notably, several results have already been found in the balanced case (Wang et. al., 2020).

Furthermore, the (entropy-regularized) Unbalanced Optimal Transport problem was studied in several papers, and Pham et. al. (2020) have established a quadratic bound on the runtime complexity of the Sinkhorn algorithm. But in their transport plan (Eq. 2 in Pham et. al., 2020), the last two parameters are equal (i.e. $\epsilon_n = \epsilon_{\theta}$). Contrastingly, in the present study, the transport plan (1) is more general. So, the results in (Pham et. al., 2020) cannot be directly applied here, and hence, some generalized convergence analysis of the Sinkhorn algorithm should be provided here.

Beyond those positioning issues, the paper has some clarity issues. In Section 2, the probability distributions $\eta$ and $\theta$ are sometimes fixed (Section 2.1), but other times they can vary in a sequential way. To this point, I would suggest first examining the GBT framework in the batch case, where  $\eta$ and $\theta$ are fixed and Eq (1) is analyzed for several scaling points. Then, this framework should be generalized to the sequential case by presenting the GBT algorithm (Algorithm 1) in Section 4, by clarifying the updates for $\theta_k$ and $\eta_k$ (note, that the update for $\eta_k$ is missing in Algorithm 1).


**Strength And Weaknesses:**

Overall, this general framework is conveying some interesting results for the study of learning agents, but it is quite difficult to assess their significance due to clarity issues and the absence of a detailed related work section (see comment below).


**Summary Of The Paper:**

This paper aims at unifying Bayesian inference, cooperative communication, and discriminative learning in a general learning framework, called Generalized Belief Transport (GBT). The cornerstone of this unifying framework lies in the paradigm of entropy-regularized unbalanced optimal transport defined using three scaling parameters, from which classical learning models can be derived using special points. The paper examines both the batch (Section 3) and the sequential (Section 4) versions of GBT, where the latter opens the door to several questions related to the convergence of online GBT for several scaling points.


**Summary Of The Review:**

Arguably, unifying Bayesian inference, communication inference, and discriminative learning is very challenging.
This paper aims at paving the way in this direction but, as indicated above, it is quite difficult to assess the significance of the results,
due to clarity issues and the lack of a detailed related work section.

---

> ### Author Response · Authors · 2022-11-19
> **Response**
>
> Thank you for your generous comments and suggestions.
> We are encouraged by your acknowledgment on the challenge we are facing on unifying the entire class learning models.
> Please see our answers to your questions below.
>
> **Q1**-A detailed related work section should be included just after the introduction.
>
> **A1**-Thank you for the suggestion, we will reorganize the text in revision.
> Meanwhile, a thorough review on existing models and how they fit in GBT are described in Section 3.2.
>
> (Wang et. al., 2020) made the connection between optimal transport and cooperative communication,
> which forms a line in the parameter space of GBT (Besides the brief intro in section 3.2, results of (Wang et. al., 2020) are detailed in Appendix A under the section Cooperative Communication).
> In addition to being significantly more general, GBT
> parameterises the entire space of learning models, which allows exploration of
> new learning approaches. Furthermore,
> the continuity of GBT (section 3.1) allows one to develop smooth interpolation between learning models.
>
>
> **Q2**-Some generalized convergence analysis of the Sinkhorn algorithm should be provided.
>
> **A2**-The general convergence of Sinkhorn algorithm for Unbalanced OT with different parameters
> on the marginals ($\eta$ and $\theta$) are proved in (Chizat et al. 2018).
> A detail treatment can be found in section 10.2 of (Peyré and Cuturi, 2019).
> Thank you for pointing out the convergence rate analysis in (Pham et. al., 2020) is only done for
> the case where the parameters on the marginals ($\eta$ and $\theta$) are the same.
> We will note it in revision.
>
> **Q3**-Regarding restructure.
>
> **A3**-Thank you for the great suggestion!
> Yes, we aim to analyze the the batch version (which we called single round) in Section 3, where $\eta$ and $\theta$ are fixed,
> and sequential version in section 4, where $\eta$ and $\theta$ are updated.
> We will emphasize this point, as you suggested, in both introduction, and beginning of each section!
> And yes, thank you for catching the missing update of $\eta_k$ in Algorithm1, have modified it in the revised draft.

---

### Official Review · Reviewer_TGqt · 2022-10-22

**Confidence:** 4
**Correctness:** 3
**Technical Novelty And Significance:** 3
**Empirical Novelty And Significance:** 2
**Recommendation:** 6

**Clarity, Quality, Novelty And Reproducibility:**

The claims and results in this paper are sound. The theoretical analysis is pretty comprehensive.

**Details Of Ethics Concerns:**

No ethic concerns.

**Strength And Weaknesses:**

Pro:
- First of all, this paper is well-written. The relationship between different learning perspectives is introduced comprehensively.
- Overall the idea of formulating such a learning problem as unbalanced optimal transport is quite novel based on some existing works connecting OT with cooperative communication.
- The authors introduced their theoretical results quite rigorously.

Cons and questions:
- In the algorithm 1, where is the $\eta$ updated? What is the data sampler $\tau$? In addition, in each iteration there will be new data samples obtained from the sampler, will that lead to a different cost matrix $C$ in each iteration?
- Maybe one limitation of this work is that, I can see its potential in unifying several ML perspectives in an elegant way however it could be great if the authors could specify some applications of the belief. It seems to me that this framework maybe is related to autoML or model selection and the authors might consider some tasks on real-world datasets.


**Summary Of The Paper:**

In this paper, the authors proposed a framework that can align a set of hypotheses models with a set of datasets. Specifically, the hypotheses and datasets are viewed as empirical distributions and certain loss functions can serve as the transport cost of moving one hypothesis to a dataset. Thus, this alignment can be reviewed as an unbalanced optimal transport problem. In the proposed formulation, the coefficients that control the KL for regularizing the marginals have a new interpretation. The authors showed that the whole learning framework is interpolating between several learning paradigms, including Bayesian inference, frequentist estimation, cooperative communication and discriminative learning. The authors also provided a detailed theoretical analysis to justify the consistency properties.

**Summary Of The Review:**

From my perspective, this draft is well presented and the proposed idea shares a new perspective on connecting several classical ML perspectives. Thus, I lean toward the acceptance of this work.

---

> ### Author Response · Authors · 2022-11-19
> **Response**
>
> Thank you for your generous comments and suggestions.We are encouraged that you find our draft is well presented and our proposed idea shares a new perspective on connecting several classical ML perspectives. Please see our answers to your questions below.
>
> **Q1**- Where is the $\eta$ updated? What is the data sampler $\tau$?
>
> **A1**- For $\eta$, thank you for pointing it out! The updating rule can be found in paragraph right above algorithm 1: assume at step $k$, data $d_i$ is observed, then the learner updates their data distribution to $\eta_k$ by increment of the $i$-th element of $\eta_{k-1}$. A line has been added in the revised draft. Regarding $\tau$, $\tau$ is a sampler that samples a data from the true hypothesis $\mathbb{P}(d|h^*)$.
> Depending on application, $\tau$ could be the environment, or a teacher who has access to the real hypothesis.
>
> **Q2**- new data samples obtained from the sampler, will that lead to a different cost matrix in each iteration?
>
> **A2**-In the current setting, we assume the cost matrix ($C$) is fixed along the process, only belief and data distribution ($\eta_k$ and $\theta_k$) are updated. However, at each step, the learner updates their estimation of the mapping $M$ between the data and hypotheses by computing an new UOT solution using $(C, \eta_k, \theta_k)$.This mapping acts in the role of likelihood as in Bayesian inference, which directly effects how the learner updates their beliefs based on the observed data.
>
> **Q3**-I can see its potential in unifying several ML perspectives in an elegant way however it could be great if the authors could specify some applications of the belief.
>
> **A3**-Regarding applications, the unification of GBT establishes a firm theoretical foundation for the entire class of learning theories,
> build upon which, basic questions in learning can be answered rigorously. For instance,
> (1) Because the entire space of learning models is parameterized, new learning model can be explored naturally (or even optimized with respect to particular tasks). A drawback of Bayesian inference is that only hypotheses in the original hypothesis set can be learned.
> Utilizing GBT, a class of learners who are able to learn *any* hypothesis is constructed and studied in section 4.
>
> (2) Interpolation between learning models can be investigated properly based on the continuity of GBT.
> Human learners appear to be capable of moving between different learning models gradually.
> Consider an individual at a carnival who is playing a game. At each of 10 trials, a bit of information is provided, but the available reward decreases. The individual has a pool of tickets with which they can bet on the outcome at each trial. The question is how the individual should update their beliefs in order to maximize their rewards. On the first trial, their belief update, in order to accurately reflect the evidence, should follow Bayes rule. However, for the last trial, one should focus bets on the most probable outcome in order to maximize chances for rewards, that is, their beliefs should be optimized for discriminating among the possible outcomes.
> GBT offers a coherent way of interpolating between these two approaches to provide candidate strategies on the intermediate steps. Such situations are common where there is an explicit constraint on the time horizon after which point no further evidence can be obtained, and there are incentives to act early, rather than to wait until evidence has fully accumulated. For example, identifying dangerous situations (tiger or not? poisonous or not?). Please see a demo on such interpolate in the last section of supplementary materials in the revised draft.
>
> Because of the the generality of GBT, there are several other cases where movement in the space of paramterizations yields interesting, novel theory. (a) Consider the case where one starts at Frequentism and moves to Bayesian inference. This represents a generalization of uninformative priors [Jeffreys H 1946] in which the prior becomes gradually informative. Note that this is different from standard minimally informative priors that accumulate evidence gradually, but always enforce the Bayesian prior. (b) Consider the case where one starts at Bayesian inference and moves to Cooperative communicaiton. This represents learning that people select data purposefully, rather than sampling at random, conditional on the hypothesis they wish to convey. (c) Consider moving from Cooperative communication to tyrant. This represents the gradual realization that the person you are communicating with is not updating their beliefs. Taken together, these demonstrate that the GBT framework allows unified reasoning about a wide array of learning phenomena, a point which we will make more clearer in the revised paper.
>
> *[Jeffreys H (1946)] "An invariant form for the prior probability in estimation problems". Proceedings of the Royal Society of London.  186 (1007): 453–461.*

---

### Official Review · Reviewer_pJWT · 2022-10-24

**Confidence:** 2
**Correctness:** 3
**Technical Novelty And Significance:** 3
**Empirical Novelty And Significance:** 2
**Recommendation:** 6

**Clarity, Quality, Novelty And Reproducibility:**

This paper is clearly written and well organized. I have not checked the details of the proofs, but the derivations seem reasonable. The conclusion that classic learning algorithms as special points on the cubic space of GBT models seem intuitive. As far as I am aware, the GBT formulation of machine learning methods is novel. Detailed proofs are provided in the appendix.

**Strength And Weaknesses:**

This paper provides a unified theoretical framework that generalizes several classic machine learning algorithms. This framework, called GBT, provides a novel perspective of these classic learning algorithms using the theory of optimal transport. The cubic view of the parameter space of GBT is intuitive, and how classic algorithms lie on this cubic space as particular points is well explained.

Overall, I found this new framework intriguing. However, it is unclear how existing methods could benefit from such an alternative perspective. Does GBT provide improvement in the classic learning settings? Does GBT permit us to learn in settings that were challenging before? It would be appreciated if the authors could provide examples to show how this novel framework permits us to improve over current baselines.

**Summary Of The Paper:**

This paper proposes a unified theoretical framework, called generalized belief transport (GBT), that generalizes subclasses of machine learning algorithms including Bayesian inference, cooperative communication, and classification. The authors visualize the parameter space of learning models encoded by GBT as a cube, with the aforementioned subclasses of classic learning algorithms as particular points. Finally, the authors investigate online learning in GBT framework, where the learner’s marginal on the data is not fixed a priori, but accumulates evidence based on experience. Asymptotic convergence properties are provided.

**Summary Of The Review:**

This paper proposes a unified theoretical framework, called generalized belief transport (GBT), that generalizes subclasses of machine learning algorithms including Bayesian inference, cooperative communication, and classification. This allows one to visualize the parameter space of learning models encoded by GBT as a cube, with the subclasses of classic learning algorithms as special points. It would be appreciated if the authors could demonstrate how the GBT framework leads to new algorithms that improve over current baselines. However, I find this theoretical framework novel and could have an impact across fields of machine learning.

---

> ### Author Response · Authors · 2022-11-19
> **Response**
>
> Thank you for your generous comments and suggestions!
> We are encouraged that you find our theoretical framework is intriguing and novel, and could have an impact across fields of machine learning.
>
> Regarding the concerns on `how existing methods could benefit from such an alternative perspective',
> GBT establishes a uniform theoretical foundation for a broad class of learning models,
> upon which, basic questions in learning can be answered rigorously.
> For instance,
>
>
> (1) Because the entire space of learning models is parameterized,
> new learning model can be explored naturally (or even optimized with respect to particular tasks).
> A drawback of Bayesian inference is that only hypotheses in the original hypothesis set can be learned.
> Utilizing GBT, a class of learners ($\epsilon_{\eta} = \epsilon_{\theta} = \infty, \epsilon_P\in (0,\infty)$) who are able to learn *any* hypothesis is
> constructed and studied in section 4.
>
> (2) Interpolation between learning models can be investigated properly based on the continuity of GBT.
> Human learners appear to be capable of moving between different learning models gradually.
> Consider an individual at a carnival who is playing a game. At each of 10 trials, a bit of information is provided, but the available reward decreases. The individual has a pool of tickets with which they can bet on the outcome at each trial. The question is how the individual should update their beliefs in order to maximize their rewards. On the first trial, their belief update, in order to accurately reflect the evidence, should follow Bayes rule. However, for the last trial, one should focus bets on the most probable outcome in order to maximize chances for rewards, that is, their beliefs should be optimized for discriminating among the possible outcomes.
> GBT offers a coherent way of interpolating between these two approaches to provide candidate strategies on the intermediate steps. Such situations are common where there is an explicit constraint on the time horizon after which point no further evidence can be obtained, and there are incentives to act early, rather than to wait until evidence has fully accumulated. For example, identifying dangerous situations (tiger or not? poisonous or not?).
> Please see a demo on such interpolate in the last section of supplementary materials in the revised draft.
>
>
> Because of the the generality of GBT, there are several other cases where movement in the space of paramterizations yields interesting, novel theory.
> (a) Consider the case where one starts at Frequentism and moves to Bayesian inference. This represents a generalization of uninformative priors [Jeffreys H (1946)] in which the prior becomes gradually informative. Note that this is different from standard minimally informative priors that accumulate evidence gradually, but always enforce the Bayesian prior.
> (b) Consider the case where one starts at Bayesian inference and moves to Cooperative communicaiton. This represents learning that people select data purposefully, rather than sampling at random, conditional on the hypothesis they wish to convey.
> (c) Consider moving from Cooperative communication to tyrant. This represents the gradual realization that the person you are communicating with is not updating their beliefs.
> Taken together, these demonstrate that the GBT framework allows unified reasoning about a wide array of learning phenomena, a point which we will make more clearer in the revised paper.
>
>
>
> *[Jeffreys H (1946)] "An invariant form for the prior probability in estimation problems".
> Proceedings of the Royal Society of London. Series A, Mathematical and Physical Sciences. 186 (1007): 453–461.*

---

> > ### Comment · Reviewer_pJWT · 2022-11-25
> > **Thanks for the response**
> >
> > I thank the authors for the provided feedback. I have read it together with the other reviewers' comments. Most of my concerns have been clarified. Still, the significance of the results is somewhat unclear due to clarity issues and the lack of detailed applications in the manuscript. I will consider adjusting my score after the discussion with the other reviewers.

---

### Official Review · Reviewer_wkW3 · 2022-10-27

**Confidence:** 4
**Correctness:** 1
**Technical Novelty And Significance:** 2
**Empirical Novelty And Significance:** 1
**Recommendation:** 1

**Clarity, Quality, Novelty And Reproducibility:**

While the contribution is novel, I'm not sure it's particularly significant. As mentioned in
the strengths and weaknesses section, the clarity of the work could be greatly improved.
Since the work has stated it's mostly theoretical and the experiment in the paper seems
straightforward, I don't have reproducibility concerns.


**Strength And Weaknesses:**

 The formalism introduced in the paper is novel, but I'm not particularly sure
 that it is all that useful. While I can see how exploring a Bayesian inference where
 the prior is made progressively more or less informative has some value, I don't
 see what benefits come from some of the other mixtures.

 I found the paper very difficult to follow. Large amounts of time is spent on
 introducing the work with fairly basic ideas being defined that likely reader
 would already be familiar with. I also am unsure some of the terminology is being
 used precisely enough. I don't know what it means to do Frequentist inference. From
 context it seems to be maximum likelihood estimation. Bayesian inference seems
 to be restricted to discrete space of parameters. None of these limitations
 are explicitly mentioned, I just don't see how the algorithm presented could
 generalize otherwise. It would be very helpful if the authors clarified precisely
 which methods their formalism can represent instead of just saying "Bayesian inference"
 and "Discriminate Learning".

The author's response did help clarify some of what these interpolations could mean, but I
remain unsure all pairs of interpolations within the cube yields an explainable learning algorithm.


**Summary Of The Paper:**

 This paper introduces a new unified formalism where Bayesian
 inference, Discriminate learning, Cooperative learning, and Maximum
 Likelihood Estimation are all expressible using a single algorithm
 inspired by the Sinkhorn algorithm from the optimal transport literature.

 The algorithm's generality is expressed with a cube where one could in
 principle express not just any of the above algorithms but linear combinations
 of them.

**Summary Of The Review:**

 This is an original idea that would greater benefit from additional clarity and better
 exposition of the value of the presented formalism.

---

> ### Author Response · Authors · 2022-11-19
> **Response**
>
> Thank you for your generous comments and suggestions.
> We are surprised, given the other reviewers' assessments, and disappointed that you find our paper hard to follow.
> Please see our clarification regarding your concerns below.
>
> **Q1** "I don't see what benefits come from some of the other mixtures."
>
> **A1** - Instead of finding a mixture of existing models, we aim to unify the entire class of learning models utilizing unbalanced optimal transport.
> Regarding the usefulness, our formulation (called GBT) establishes a firm uniformed theoretical foundation for the learning theory,
> built upon which, basic questions in learning can be answered rigorously.
> For instance,
>
> (1) Because the entire space of learning models is parameterized,
> new learning model can be explored naturally (or even optimized with respect to particular tasks).
> A drawback of Bayesian inference is that only hypotheses in the original hypothesis set can be learned.
> Utilizing GBT, a class of learners ($\epsilon_{\eta} = \epsilon_{\theta} = \infty, \epsilon_P\in (0,\infty)$) who are able to learn *any* hypothesis is
> constructed and studied in section 4.
>
> (2) Interpolation between learning models can be investigated properly based on the continuity of GBT.
> Human learners appear to be capable of moving between different learning models smoothly.
>
> Consider an individual at a carnival who is playing a game. At each of 10 trials, a bit of information is provided, but the available reward decreases. The individual has a pool of tickets with which they can bet on the outcome at each trial. The question is how the individual should update their beliefs in order to maximize their rewards. On the first trial, their belief update, in order to accurately reflect the evidence, should follow Bayes rule. However, for the last trial, one should focus bets on the most probable outcome in order to maximize chances for rewards, that is, their beliefs should be optimized for discriminating among the possible outcomes.
> GBT offers a coherent way of interpolating between these two approaches to provide candidate strategies on the intermediate steps. Such situations are common where there is an explicit constraint on the time horizon after which point no further evidence can be obtained, and there are incentives to act early, rather than to wait until evidence has fully accumulated. For example, identifying dangerous situations (tiger or not? poisonous or not?).
> Please see a demo on such interpolate in the last section of supplementary materials in the revised draft.
>
> Because of the the generality of GBT, there are several other cases where movement in the space of paramterizations yields interesting, novel theory.
> (a) Consider the case where one starts at Frequentism and moves to Bayesian inference. This represents a generalization of uninformative priors [Jeffreys H (1946)] in which the prior becomes gradually informative. Note that this is different from standard minimally informative priors that accumulate evidence gradually, but always enforce the Bayesian prior.
> (b) Consider the case where one starts at Bayesian inference and moves to Cooperative communicaiton. This represents learning that people select data purposefully, rather than sampling at random, conditional on the hypothesis they wish to convey.
> (c) Consider moving from Cooperative communication to tyrant. This represents the gradual realization that the person you are communicating with is not updating their beliefs.
> Taken together, these demonstrate that the GBT framework allows unified reasoning about a wide array of learning phenomena, a point which we will make more clearer in the revised paper.
>
>
> *[Jeffreys H (1946)] "An invariant form for the prior probability in estimation problems".
> Proceedings of the Royal Society of London. Series A, Mathematical and Physical Sciences. 186 (1007): 453–461.*
>
> **Q2**-Large amounts of time is spent on introducing the work with fairly basic ideas.
>
> **A2**-Brief review are included to explain how existing models fit into the GBT framework naturally.
>
> **Q3**- What is Frequentist inference.
>
> **A3**- Frequentist inference refers to a learner who updates their belief according to the observation frequency,
> not maximal likelihood. In particular, in our context, no probabilistic model need be assumed.

---

### Decision · Program_Chairs · 2023-01-20

**Decision:**

Reject

**Justification For Why Not Higher Score:**

The work is not well-motivated and the lack of clarity makes it difficult to have reasonable assessment of the work.

**Justification For Why Not Lower Score:**

N/A

**Metareview: Summary, Strengths And Weaknesses:**

Although the reviewers see novel ideas in this work, they are seriously concerned about the importance/applications/benefits of the setting studied in the paper. There are also major concerns about the writing of the paper, its structure, the lack of related work, and the lack of clarity and preciseness. The last issue, the lack of clarity, in particular, makes it difficult for the reviewers to have a proper assessment of this work. I would strongly recommend that the authors take the reviewers' comments into account, undertake a major revision of their work, and prepare it for a future conference.